# Gaussian Mixture Models Based Augmentation Enhances GNN Generalization

## Abstract

Graph Neural Networks (GNNs) have shown great promise in tasks like node and graph classification, but they often struggle to generalize, particularly to unseen or out-of-distribution (OOD) data. These challenges are exacerbated when training data is limited in size or diversity. To address these issues, we introduce a theoretical framework using Rademacher complexity to compute a regret bound on the generalization error and then characterize the effect of data augmentation. This framework informs the design of GMM-GDA, an efficient graph data augmentation (GDA) algorithm leveraging the capability of Gaussian Mixture Models (GMMs) to approximate any distribution. Our approach not only outperforms existing augmentation techniques in terms of generalization but also offers improved time complexity, making it highly suitable for real-world applications.

## 1 Introduction

Graphs are a fundamental and ubiquitous structure for modeling complex relationships and interactions. In biology, graphs are employed to represent complex networks of protein interactions and in drug discovery by modeling molecular relationships. Similarly, in social networks, graphs capture relationships and community interactions, offering insights into social structures and interactions (Zeng et al., 2022; Gaudelet et al., 2021; Newman et al., 2002). To address the unique challenges posed by graph-structured data, GNNs have been developed as a specialized class of neural networks designed to operate directly on graphs. Unlike traditional neural networks that are optimized for grid-like data, such as images or sequences, GNNs are engineered to process and learn from the relational information embedded in graph structures. GNNs have demonstrated state-of-the-art performance across a range of graph representation learning tasks such as node and graph classification, proving their effectiveness in various real-world applications (Vignac et al., 2022; Corso et al., 2022; Duval et al., 2023; Castro-Correa et al., 2024; Chi et al., 2022).

Despite their impressive capabilities, GNNs face significant challenges related to generalization, particularly when handling unseen or out-of-distribution (OOD) data (Guo et al., 2024; Li et al., 2022). OOD graphs are those that differ significantly from the training data in terms of graph structure, node features, or edge types, making it difficult for GNNs to adapt and perform well on such data. This challenge is also faced when GNNs are trained on small datasets, where the limited data diversity hampers the model's ability to generalize effectively. To address these challenges, the community has explored various strategies to improve the robustness and generalization ability of GNNs (Abbahaddou et al., 2024; Yang et al., 2022). One promising approach is data augmentation, which involves artificially expanding the training dataset by introducing variations of the original graph data. Data augmentation has shown its benefits across different types of data structures such as images (Krizhevsky et al., 2012) and time series (Aboussalah et al., 2023). For graph data structures, generating augmented versions of the original graphs, such as by adding or removing nodes and edges or perturbing node features (Rong et al., 2019; You et al., 2020), allows for the creation of a more varied training set. Inspired by the success of the Mixup technique in computer vision (Rebuffi et al., 2021; Dabouei et al., 2021; Hong et al., 2021), additional methods such as $\mathcal{G}$-Mixup and GEOMIX have been developed to adapt the Mixup technique for graph data (Ling et al., 2023; Han et al., 2022). These techniques combine different graphs to create new, synthetic training examples, further enriching the dataset and enhancing the GNN's ability to generalize to new unseen graph structures.

In this work, we introduce a novel graph augmentation technique based on Gaussian Mixture Models (GMMs), which operates at the level of the final hidden representations. Specifically, guided by our theoretical results, we apply the Expectation-Maximization (EM) algorithm to train a GMM on the graph representations. We then use this GMM to generate new augmented graph representations through sampling, enhancing the diversity of the training data.

**Contributions.** The contributions of our work are as follows:

- **Theoretical framework for generalization in GNNs:** We introduce a theoretical framework that rigorously analyzes how graph data augmentation impacts the generalization capabilities of GNNs. This framework offers new insights into the underlying mechanisms that drive performance improvements through augmentation.
- **Efficient graph data augmentation via GMMs:** We propose GMM-GDA, a fast and efficient graph data augmentation technique, leveraging GMMs. This approach enhances the diversity of training data while maintaining computational simplicity, making it scalable for large graph datasets.
- **Comprehensive theoretical analysis using influence functions:** We perform an in-depth theoretical analysis of our augmentation strategy through the lens of influence functions, providing a principled understanding of the approach's impact on generalization performance.

## 2 BACKGROUND AND RELATED WORK

**Graph Neural Networks.** Let $\mathcal{G} = (\mathcal{V}, \mathcal{E})$ denote a graph, where $\mathcal{V}$ represents the set of vertices and $\mathcal{E}$ represents the set of edges. We use $L = |\mathcal{V}|$ to denote the number of vertices and $m = |\mathcal{E}|$ to denote the number of edges. For a node $v \in V$, let $\mathcal{N}(v)$ be the set of its neighbors, defined as $\mathcal{N}(v) = \{u \colon (v, u) \in E\}$. The degree of vertex $v$ is the number of neighbors it has, which is $|\mathcal{N}(v)|$. A graph is commonly represented by its adjacency matrix $\mathbf{A} \in \mathbb{R}^{L \times L}$, where the $(i, j)$-th element of this matrix is equal to the weight of the edge between the $i$-th and $j$-th node of the graph and a weight of zero in case the edge does not exist. Additionally, in some cases, nodes may have associated feature vectors. We denote these node features by $\mathbf{X} \in \mathbb{R}^{L \times D}$ where $D$ is the dimensionality of the features.

A GNN model consists of multiple neighborhood aggregation layers that use the graph structure and the feature vectors from the previous layer to generate updated representations for the nodes. Specifically, GNNs update a node's feature vector by aggregating information from its local neighborhood. Consider a GNN model with $T$ neighborhood aggregation layers. Let $\mathbf{h}_v^{(0)}$ denote the initial feature vector of node $v$, which is the corresponding row in $\mathbf{X}$. At each layer $t > 0$, the hidden state $\mathbf{h}_v^{(t)}$ of node $v$ is updated as follows:

$$\mathbf{a}_v^{(t)} = \text{AGGREGATE}^{(t)}\Big( \big\{ \mathbf{h}_u^{(t-1)} \colon u \in \mathcal{N}(v) \big\} \Big),$$

$$\mathbf{h}_v^{(t)} = \text{COMBINE}^{(t)}\Big( \mathbf{h}_v^{(t-1)}, \mathbf{a}_v^{(t)} \Big),$$

where $\text{AGGREGATE}(\cdot)$ is a permutation-invariant function that combines the feature vectors of $v$'s neighbors into an aggregated vector. This aggregated vector, together with the previous feature vector $\mathbf{h}_v^{(t-1)}$, is fed to the $\text{COMBINE}(\cdot)$ function, which merges these two vectors to produce the updated feature vector of $v$. Two popular GNN architechtures are Graph Convolution Networks (GCN) and Graph Isomorphism Networks (GIN) (Kipf & Welling, 2017; Xu et al., 2019).

After $T$ iterations of neighborhood aggregation, to produce a graph-level representation, GNNs apply a permutation invariant readout function, e.g., the sum operator, to nodes feature as follows:

$$\mathbf{h}_\mathcal{G} = \text{READOUT}\Big( \big\{ \mathbf{h}_v^{(T)} \colon v \in V \big\} \Big). \tag{1}$$

**Data Augmentation for Graphs.** Graph data augmentation has become an essential aspect of enhancing the performance and robustness of GNNs. Among the classical techniques, structural modifications of the graph are widely used to generate augmented training graphs. Key methods in this category include DropEdge, DropNode, and Subgraph sampling techniques (Rong et al., 2019;

You et al., 2020). For example, the DropEdge technique randomly removes a subset of edges from the graph during training, improving the model's robustness to missing or noisy connections. Similarly, DropNode removes certain nodes as well as their connections, assuming that the missing part of nodes will not affect the semantic meaning, i.e., the structural and relational information of the original graph. Another method is Subgraph, which samples a subgraph from the original graph using random walk to use as a training graph. By training on these augmented graphs, GNNs can generalize to unseen graph structures more efficiently.

Beyond classical methods, recent advancements have explored more sophisticated augmentation techniques, focusing on manipulating graph embeddings and leveraging geometric properties of graphs. Following the effectiveness of the Mixup technique in computer vision (Rebuffi et al., 2021; Dabouei et al., 2021; Hong et al., 2021), several works describe variations of the Mixup for graphs. For example, the Manifold-Mixup model conducts a Mixup operation for graph classification in the embedding space. This technique interpolates between graph-level embeddings after the READOUT function, blending different graphs in the embedding space (Wang et al., 2021). $\mathcal{G}$-Mixup (Han et al., 2022) uses graphons to model the topological structures of each graph class and then interpolates the graphons of different classes, subsequently generating synthetic graphs by sampling from mixed graphons across different classes. It is important to note that $\mathcal{G}$-Mixup operates under a significant assumption: graphs belonging to the same class can be produced by a single graphon. The $S$-Mixup method, for a given pair of graphs, determines node-level correspondences between the nodes in both graphs and subsequently interpolates the graphs (Ling et al., 2023). FGW-Mixup adopts the Fused Gromov–Wasserstein barycenter as the mixup graphs, but suffers from heavy computation (Ma et al., 2024). Finally, the GeoMix technique (Zeng et al., 2024) uses mixup graphs on the exact Gromov-Wasserstein geodesics.

**Gaussian Mixture Models.** GMMs are probabilistic models used for modeling complex data by representing them as a mixture of multiple Gaussian distributions. The probability density function $p(\mathbf{x})$ of a data point $\mathbf{x}$ in a GMM with $K$ Gaussian components is given by:

$$p(\mathbf{x}) = \sum_{k=1}^{K} \pi_k \mathcal{N}(\mathbf{x} \mid \boldsymbol{\mu}_k, \boldsymbol{\Sigma}_k), \tag{2}$$

where $\pi_k$ is the weight of the $k$-th Gaussian component, with $\pi_k \geq 0$ and $\sum_{k=1}^{K} \pi_k = 1$, and $\mathcal{N}(\mathbf{x} \mid \boldsymbol{\mu}_k, \boldsymbol{\Sigma}_k)$ is the Gaussian probability density function for the $k$-th component, defined as:

$$\mathcal{N}(\mathbf{x} \mid \boldsymbol{\mu}_k, \boldsymbol{\Sigma}_k) = \frac{1}{(2\pi)^{d/2} \det(\boldsymbol{\Sigma}_k)^{1/2}} \exp\left(-\frac{1}{2}(\mathbf{x} - \boldsymbol{\mu}_k)^{\top} \boldsymbol{\Sigma}_k^{-1} (\mathbf{x} - \boldsymbol{\mu}_k)\right),$$

where $\boldsymbol{\mu}_k$ and $\boldsymbol{\Sigma}_k$ are respectively the mean vectors and the covariance vectors of the $k$-th Gaussian component, and $d$ the dimensionality of $\mathbf{x}$. The parameters of a GMM are typically estimated using the EM algorithm (Dempster et al., 1977), which alternates between estimating the membership probabilities of data points for each Gaussian component (Expectation step) and updating the parameters of the Gaussian distributions (Maximization step). GMMs are a powerful tool in statistics and machine learning and are used for various purposes, including clustering and density estimation (Ozertem & Erdogmus, 2011; Naim & Gildea, 2012; Zhang et al., 2021).

## 3 GMM-GDA: GAUSSIAN MIXTURE MODEL FOR GRAPH DATA AUGMENTATION

In this section, we begin by introducing the mathematical framework for graph data augmentation and its connection to the generalization of GNNs. Following that, we present our proposed model GMM-GDA, which is based on GMMs for graph augmentation.

### 3.1 MATHEMATICAL FORMALISM

We focus on the task of graph classification, where the objective is to classify graphs into predefined categories. Given a training dataset of graphs $\mathcal{D}_{\text{train}} = \{(\mathcal{G}_n, y_n) \mid n = 1, \ldots, N\}$, $\mathcal{G}_n$ is the $n$-th graph and $y_n$ is its corresponding label belonging to a set $\{0, \ldots, C\}$. Each graph $\mathcal{G}_n$ is represented as a tuple $(\mathcal{V}_n, \mathcal{E}_n, \mathbf{X}_n)$, where $\mathcal{V}_n$ denotes the set of nodes with cardinality $L_n = |\mathcal{V}_n|$, $\mathcal{E}_n \subseteq \mathcal{V}_n \times \mathcal{V}_n$

is the set of edges, and $\mathbf{X}_n \in \mathbb{R}^{L_n \times D}$ is the node feature matrix of dimension $D$. The objective is to train a GNN $f(\cdot, \theta)$ that can accurately predict the class labels for unseen graphs in the test set $\mathcal{D}_{\text{test}} = \{\mathcal{G}_n^{test} \mid n = 1, \ldots, N_{\text{test}}\}$. The classical training approach involves minimizing the following loss function,

$$\mathcal{L} = \frac{1}{N} \sum_{n=1}^{N} \ell(f(\mathcal{G}_n, \theta), y_n), \tag{3}$$

where $\ell$ denotes the cross-entropy loss function. To improve the robustness and generalization ability of the GNN, we introduce data augmentation for graphs. For each graph $\mathcal{G}_n$, we generate an augmented graph $\mathcal{G}_n^\lambda$ using a data augmentation strategy $A_\lambda$, where $\lambda$ is a parameter sampled from a prior distribution $\mathcal{P}$, such as a uniform distribution. The data augmentation strategy $A_\lambda$ is defined as a function: $A_\lambda : \mathcal{G}_n \in G \to \mathcal{G}_n^\lambda = A(\mathcal{G}_n, \lambda) \in G$, where $G$ is the set of all possible graphs with $n$ nodes. With the augmented data, the loss function is modified to account for multiple augmented versions of each graph:

$$\mathcal{L}^{\text{aug}} = \frac{1}{N} \sum_{n=1}^{N} \mathbb{E}_{\lambda \sim \mathcal{P}} \left[ \ell(f(\mathcal{G}_n^\lambda, \theta), y_n) \right]. \tag{4}$$

For simplicity, we denote the loss for the original graph as $\ell(f(\mathcal{G}_n, \theta), y_n) = \ell(\mathcal{G}_n, \theta)$ and the loss for an augmented graph as $\mathbb{E}_{\lambda \sim \mathcal{P}} \left[ \ell(f(\mathcal{G}_n^\lambda, \theta), y_n) \right] = \ell_{aug}(\mathcal{G}_n, \theta)$. The loss $\ell_{aug}$ is empirically estimated as follows,

$$\ell_{aug}(\mathcal{G}_n, \theta) = \mathbb{E}_{\lambda \sim \mathcal{P}} \left[ \ell(f(\mathcal{G}_n^\lambda, \theta), y_n) \right] \simeq \frac{1}{M} \sum_{m=1}^{M} \ell(f(\mathcal{G}_n^{\lambda_{n,m}}, \theta), y_n), \tag{5}$$

where $M$ denotes the number of augmented samples per graph and $\{\lambda_{n,m}\}_{m=1}^{M}$ are the parameters sampled from $\mathcal{P}$ for each training graph $\mathcal{G}_n$. To understand the impact of data augmentation on the graph classification performance, we analyze the effect of sampling strategy $\mathcal{P}$ on the generalization risk $\mathbb{E}_{\mathcal{G} \sim G} \left[ \ell(\mathcal{G}, \theta) \right]$. More specifically, we want to study the generalization error $\eta = \mathbb{E}_{\mathcal{G} \sim G} \left[ \ell(\mathcal{G}, \theta_{aug}) \right] - \mathbb{E}_{\mathcal{G} \sim G} \left[ \ell(\mathcal{G}, \theta_\star) \right]$, where $\theta_{aug}$ and $\theta_\star$ are the optimal GNN parameters for the augmented and non-augmented settings,

$$\theta_\star = \arg\min_\theta \mathbb{E}_{\mathcal{G} \sim G} \left[ \ell(\mathcal{G}, \theta) \right], \theta_{aug} = \arg\min_\theta \mathbb{E}_{\mathcal{G} \sim G} \left[ \ell_{aug}(\mathcal{G}, \theta) \right] = \arg\min_\theta \mathbb{E}_{\mathcal{G} \sim G} \mathbb{E}_{\lambda \sim \mathcal{P}} \left[ \ell(\mathcal{G}^\lambda, \theta) \right],$$

and which can be estimated empirically as follows,

$$\hat{\theta} = \arg\min_\theta \frac{1}{N} \sum_{n=1}^{N} \ell(\mathcal{G}_n, \theta),$$

$$\hat{\theta}_{aug} = \arg\min_\theta \frac{1}{N} \sum_{n=1}^{N} \mathbb{E}_{\lambda \sim \mathcal{P}} \left[ \ell(\mathcal{G}_n^\lambda, \theta) \right] \simeq \arg\min_\theta \frac{1}{N \times M} \sum_{n=1}^{N} \sum_{m=1}^{M} \ell(\mathcal{G}_n^{\lambda_{n,m}}, \theta).$$

By theoretically studying the generalization error $\eta$, we aim to quantify the effect of each augmentation strategy on the overall classification performance, providing insights into the benefits and potential trade-offs of data augmentation in graph-based learning tasks. In Theorem 3.1, we present a regret bound of the generalization error using Rademacher complexity defined as follows Yin et al. (2019),

$$\mathcal{R}(\ell) = \mathbb{E}_{\epsilon_n \sim P_\epsilon} \left[ \sup_{\theta \in \Theta} \left| \frac{1}{N} \sum_{n=1}^{N} \epsilon_n \ell(\mathcal{G}_n, \theta) \right| \right],$$

where $\epsilon_n$ are independent Rademacher variables, taking values $+1$ or $-1$ with equal probability, $P_\epsilon$ is the Rademacher distribution, and $\Theta$ is the hypothesis class. Rademacher complexity is a fundamental concept in statistical learning which indicates how well a learned function will perform on unseen data (Shalev-Shwartz & Ben-David, 2014). Lower Rademacher complexity indicates a better generalization.

**Theorem 3.1.** *Let $\ell$ be a classification loss function with $L_{Lip}$ as a Lipschitz constant such as $\ell(\cdot, \cdot) \in [0, 1]$. Then, with a probability at least $1 - \delta$ over the samples $\mathcal{D}_{train}$, we have,*

$$\mathbb{E}_{\mathcal{G} \sim G} \left[ \ell(\mathcal{G}, \hat{\theta}_{aug}) \right] - \mathbb{E}_{\mathcal{G} \sim G} \left[ \ell(\mathcal{G}, \theta_\star) \right] \leq 2\mathcal{R}(\ell_{aug}) + 5\sqrt{\frac{2\log(4/\delta)}{N}} + 2L_{Lip} \mathbb{E}_{\mathcal{G} \sim G} \mathbb{E}_{\lambda \sim \mathcal{P}} \left[ \left\| \mathcal{G}^\lambda - \mathcal{G} \right\| \right].$$

*Moreover, we have,*

$$\mathcal{R}(\ell_{aug}) \leq \mathcal{R}(\ell) + \max_{n \in \{1,...,N\}} L_{Lip} \mathbb{E}_{\lambda \sim \mathcal{P}} \left[ \left\| \mathcal{G}_n^\lambda - \mathcal{G}_n \right\| \right].$$

Theorem 3.1 relies on the assumption that the loss function is Lipschitz continuous. This assumption is realistic given that the input node features and graph structures in real-world datasets are typically bounded. Additionally, we can ensure that the loss function is bounded within $[0, 1]$ by composing any standard classification loss with a strictly increasing function that maps values to the interval $[0, 1]$. A direct implication of Theorem 3.1 is that if we chose the right data augmentation strategy $A_\lambda$ that minimizes the expected distance between original graphs and augmented ones $\mathbb{E}_{\mathcal{G} \sim G} \mathbb{E}_{\lambda \sim \mathcal{P}} \left[ \left\| \mathcal{G}^\lambda - \mathcal{G} \right\| \right]$, we can guarantee with a high probability that the data augmentation decreases both the Rademacher complexity and the generalization risk. On the other hand, if the distance is large, we cannot guarantee that data augmentation will outperform the normal training setting.

The findings of Theorem 3.1 hold for all norms defined on the graph input space. Specifically, let us consider the graph space $(\mathbb{G}, \|\cdot\|_{\mathbb{G}})$ and the feature space $(\mathbb{X}, \|\cdot\|_{\mathbb{X}})$, where $\|\cdot\|_{\mathbb{G}}$ and $\|\cdot\|_{\mathbb{X}}$ denote the norms applied to the graph structure and features, respectively. Assuming a maximum number of nodes per graph, which is a realistic assumption for real-world data, the product space $\mathbb{G} \times \mathbb{X}$ is a finite-dimensional real vector space, and all the norms are equivalent. Thus, the choice of norm does not affect the theorem, as long as the Lipschitz constant is adjusted accordingly. Additional details and insights on the graph distance metrics can be found in Appendix G.

## 3.2 PROPOSED APPROACH

Based on the theoretical findings, it is crucial to employ a data augmentation technique that effectively controls the term $\mathbb{E}_{\mathcal{G} \sim G} \mathbb{E}_{\lambda \sim \mathcal{P}} \left[ \left\| \mathcal{G}^\lambda - \mathcal{G} \right\| \right]$, to achieve stronger generalization guarantees. This consideration leads us to explore universal approximators, particularly GMMs, which are well-suited for this purpose, and can effectively approximate any data distribution, c.f. Theorem 3.2.

**Theorem 3.2.** *(Goodfellow et al., 2016), Page 65. A Gaussian mixture model is a universal approximator of densities, in the sense that any smooth density can be approximated with any specific nonzero amount of error by a Gaussian mixture model with enough components.*

To achieve this, we first train a standard GNN on the graph classification task using the training set. Next, we obtain embeddings for all training graphs using the READOUT output, resulting in $\mathcal{H} = \{h_{\mathcal{G}_n} \ s.t. \ \mathcal{G}_n \in \mathcal{D}_{train}\}$. These embeddings are used as the basis for generating augmented training graphs. We then partition the training set $\mathcal{D}_{train}$ by classes, such that $\mathcal{D}_{train} = \bigcup_c \mathcal{D}_c$ where $\mathcal{D}_c = \{\mathcal{G}_n \in \mathcal{D}_{train} \ , \ y_n = c\}$. The objective is to learn new graph representations from these embeddings, and create augmented data for improved training.

We use the EM algorithm to learn the best-fitting GMM for the embeddings of each cluster $\mathcal{D}_c$, denoted as $\mathcal{H}_c = \{\mathbf{h}_{\mathcal{G}_n} \ s.t. \ \mathcal{G}_n \in \mathcal{D}_c\}$. The EM algorithm finds maximum likelihood estimates for each cluster $\mathcal{H}_c$. We first initialize the GMM distribution as in Eq. 2. Given a number of Gaussian distributions $K$, we specifically initialize the mean vector $\mu_k$, the covariance vector $\Sigma_k$, and the weight $\pi_k$ of each Gaussian distribution. The process then evolves iteratively: *(i)* Evaluate the posterior probabilities $\{\gamma_{ik}\}_{i,k}$, using the values of the mean vectors and covariance matrix (E-Step) Watanabe et al. (2010).

$$\gamma_{ik} = \frac{\pi_k \mathcal{N}(x_i \mid \boldsymbol{\mu}_k, \boldsymbol{\Sigma}_k)}{\sum_{j=1}^{K} \pi_j \mathcal{N}(x_i \mid \boldsymbol{\mu}_j, \boldsymbol{\Sigma}_j)}.$$

*(ii)* Estimate new parameters $\boldsymbol{\mu}_k$, $\boldsymbol{\Sigma}_k$ and $\pi_k$ with the updated values of $\{\gamma_{ik}\}_{i,k}$ (M-Step),

$$\boldsymbol{\mu}_k = \frac{\sum_{i=1}^{N} \gamma_{ik} x_i}{\sum_{i=1}^{N} \gamma_{ik}}, \quad \pi_k = \frac{1}{N} \sum_{i=1}^{N} \gamma_{ik},$$

$$\boldsymbol{\Sigma}_k = \frac{\sum_{i=1}^{N} \gamma_{ik} (x_i - \boldsymbol{\mu}_k)(x_i - \boldsymbol{\mu}_k)^\top}{\sum_{i=1}^{N} \gamma_{ik}}.$$

Once a GMM distribution $p_c$ is fitted for each cluster $\mathcal{D}_c$, we use this GMM to generate new augmented data by sampling hidden representations from $p_c$. Each new sample drawn from $p_c$ is

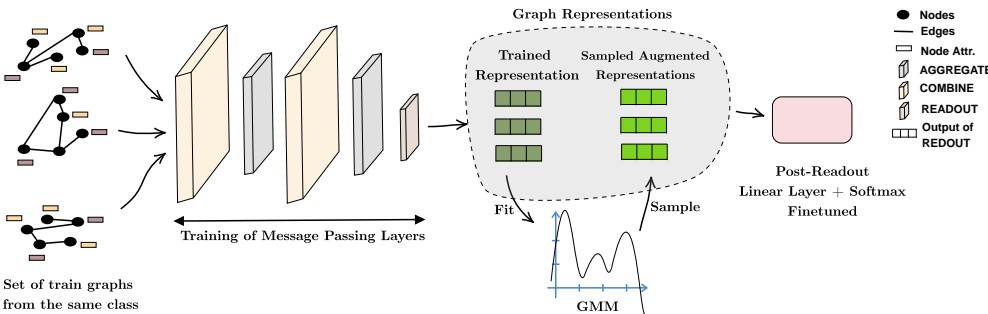

Figure 1: Illustration of GMM-GDA: Step 1. We first train the GNN on the graph classification task using the training graphs. Step 2. Next, we utilize the weights from the message passing layers to generate graph representations for the training graphs. Step 3. A GMM is then fit to these graph representations, from which we sample new graph representations. Step 4. Finally, we fine-tune the post-readout function for the graph classification task, using both the original training graphs and the augmented graph representations. For inference on the test set, we use the message passing weights trained in Step 1 and the post-readout function weights trained in Step 4.

then assigned the corresponding cluster label $c$, ensuring that the augmented data inherits the label structure from the original clusters. After merging the hidden representations of both the original training data and the augmented graph data, we finetune the post-readout function, i.e., the final part of the GNN, which occurs after the readout function, on the graph classification task. Since the post-readout function consists of a linear layer followed by a Softmax function, the finetuning process is relatively fast. To evaluate our model during inference on test graphs, we input the test graphs into the GNN layers trained in the initial step to compute the hidden graph representations. For the post-readout function, we use the weights obtained from the second stage of training. Algorithm 1 and Figure 1 provide a summary of this approach.

---

**Algorithm 1:** Detailed Steps in the GMM-GDA Algorithm

---

**Inputs:** GNN of $T$ layers $f(\cdot, \theta) = \Psi \circ \text{READOUT}\left(\cup_{t=0}^{T}\{\text{AGGREGATE}^{(t)} \circ \text{COMBINE}^{(t)}(\cdot)\}\right)$

where $\Psi$ is the post-readout function, Graph classification dataset $\mathcal{D}$, Loss function $\mathcal{L}$,

**Steps:**

  **1.** Train the GNN $f$ on the graph classification task on the training set $\mathcal{D}_{train}$;

  **2.** Use the trained Message Passing layers and the readout function to generate graph representation
      $\mathcal{H} = \{\mathbf{h}_{\mathcal{G}_n} \ s.t. \ \mathcal{G}_n \in \mathcal{D}_{train}\}$ for the training set;

  **3.** Partition the training set $\mathcal{D}_{train}$ by classes, such that $\mathcal{D}_{train} = \bigcup_c \mathcal{D}_c$ where
      $\mathcal{D}_c = \{\mathcal{G}_n \in \mathcal{D}_{train} \ , \ y_n = c\}$;

**foreach** $c \in \{0, \ldots, C\}$ **do**

    **3.1.** Fit a GMM distribution $p_c$ on the graph representations $\mathcal{H}_c = \{\mathbf{h}_{\mathcal{G}_n} \ s.t. \ \mathcal{G}_n \in \mathcal{D}_c\}$;

    **3.2.** Sample new graph representation $\widetilde{\mathcal{H}_c} = \{\widetilde{\mathbf{h}} \ s.t. \ \widetilde{\mathbf{h}} \sim p_c\}$ from the distribution $p_c$;

    **3.3.** Include the sampled representations $\widetilde{\mathcal{H}_c}$ with trained representations $\mathcal{H}_c = \mathcal{H}_c \cup \widetilde{\mathcal{H}_c}$;

**end foreach**

  **4.** Finetune the post-Readout function $\Psi$ on the graph classification task directly on
      the new training set $\mathcal{H} = \cup_c \mathcal{H}_c$.

---

## 3.3 TIME COMPLEXITY

One advantage of our approach is its efficiency, as it generates new augmented graph representations with minimal computational time. Unlike baseline methods, which apply augmentation strategies to each individual training graph (or pair of graphs in Mixup-based approaches) separately, our method learns the distribution of graph representations across the entire training dataset simultaneously using the EM algorithm (Ng, 2000). If $N = |\mathcal{D}_{train}|$ is the number of training graphs in the dataset, $d$ is the dimension of graph hidden representations $\{\mathbf{h}_{\mathcal{G}}, \ \mathcal{G} \in \mathcal{D}_{train}\}$, and $K$ is the number of Gaussian Components in the GMM, then the complexity to fit a GMM on $T$ iterations is $\mathcal{O}(N \cdot K \cdot T \cdot d^2)$ (Yang

et al., 2012). We compare the data augmentation times of our approach and the baselines in Table 7. Due to our different training scheme, i.e., where we first train the message passing layers and then train the pooling function after learning the GMM distribution, we measured the total backpropagation time and compared it with the backpropagation time of the baseline methods. The training time of baseline models varies depending on the augmentation strategy used, specifically, whether it involves pairs of graphs or individual graphs. Even in cases where a graph augmentation has a low computational cost for some baselines, training can still be time consuming as multiple augmented graphs are required to achieve satisfactory test accuracy. In contrast, GMM-GDA generates only one augmented graph per training graph, demonstrating effective generalization on the test set. Overall, our data augmentation approach is highly efficient during the sampling of augmented data, with minimal impact on the overall training time.

### 3.4 Analyzing the Generalization Ability of the Augmented Graphs via Influence Functions

We used *influence functions* (Law, 1986; Koh & Liang, 2017; Kong et al., 2021) to understand the impact of augmented data on the model performance on the test set, and thus motivate the use of data augmentation strategy which is specific to the model architecture and the model weights. In Theorem 3.3, we derive a closed-formula for the impact of adding an augmented graph $\mathcal{G}_n^{\lambda_{n,m}}$ on the GNN's performance on a test graph $\mathcal{G}_k^{test}$, where the GNN is trained solely on the original training set, without including the augmented graph.

**Theorem 3.3.** *Given a test graph $\mathcal{G}_k$ from the test set, let $\hat{\theta} = \arg\min_\theta \mathcal{L}$ be the GNN parameters that minimize the objective function in Eq. 3. The impact of upweighting the objective function $\mathcal{L}$ to $\mathcal{L}_{n,m}^{aug} = \mathcal{L} + \epsilon_{n,m}\ell(\mathcal{G}_n^{\lambda_{n,m}}, \theta)$, where $\mathcal{G}_n^{\lambda_{n,m}}$ is an augmented graph candidate of the training graph $\mathcal{G}_n$ and $\epsilon_{n,m}$ is a sufficiently small perturbation parameter, on the model performance on the test graph $\mathcal{G}_k^{test}$ is given by*

$$\frac{d\ell(\mathcal{G}_k^{test}, \hat{\theta}_{\epsilon_{n,m}})}{d\epsilon_{n,m}} = -\nabla_\theta \ell(\mathcal{G}_k^{test}, \hat{\theta}) H_{\hat{\theta}}^{-1} \nabla_\theta \ell(\mathcal{G}_n^{\lambda_{n,m}}, \hat{\theta}),$$

*where $\hat{\theta}_{\epsilon_{n,m}} = \arg\min_\theta \mathcal{L}_{n,m}^{aug}$ denotes the parameters that minimize the upweighted objective function $\mathcal{L}_{n,m}^{aug}$ and $H_{\hat{\theta}} = \nabla_\theta^2 \mathcal{L}(\hat{\theta})$ is the Hessian Matrix of the loss w.r.t the model parameters.*

We provide the proof of Theorem 3.3 in Appendix B. The influence scores are useful for evaluating the effectiveness of the augmented data on each test graph. The strength of influence function theory lies in its ability to analyze the effect of adding augmented data to the training set without actually retraining on this data. As noticed, these influence scores depend not only on the augmented graphs themselves, but also on the model's weights and architecture. This highlights the need for a graph data augmentation strategy tailored specifically to the GNN backbone in use, as opposed to traditional, techniques like DropNode, DropEdge, and $\mathcal{G}$-Mixup, which are general-purpose methods that can be applied with any GNN architecture.

We can measure the average influence $\mathcal{I}(\mathcal{G}_n^{\lambda_{n,m}})$ of a augmented graph $\mathcal{G}_n^{\lambda_{n,m}}$ on the whole test set by averaging the derivatives as follows,

$$\mathcal{I}(\mathcal{G}_n^{\lambda_{n,m}}) = \frac{-1}{|\mathcal{D}_{test}|} \sum_{\mathcal{G}_k^{test} \in \mathcal{D}_{test}} \frac{d\ell(\mathcal{G}_k^{test}, \hat{\theta}_{\epsilon_{n,m}})}{d\epsilon_{n,m}}.$$

A negative value of $\mathcal{I}(\mathcal{G}_n^{\lambda_{n,m}})$ indicates that adding the augmented data to the training set would increase the prediction loss on the test set, negatively affecting the GNN's generalization. In contrast, a good augmented graph is one with a postive $\mathcal{I}(\mathcal{G}_n^{\lambda_{n,m}})$, indicating improved generalization. In Figure 2, we present the density of the average influence scores of each augmented data on the test set.

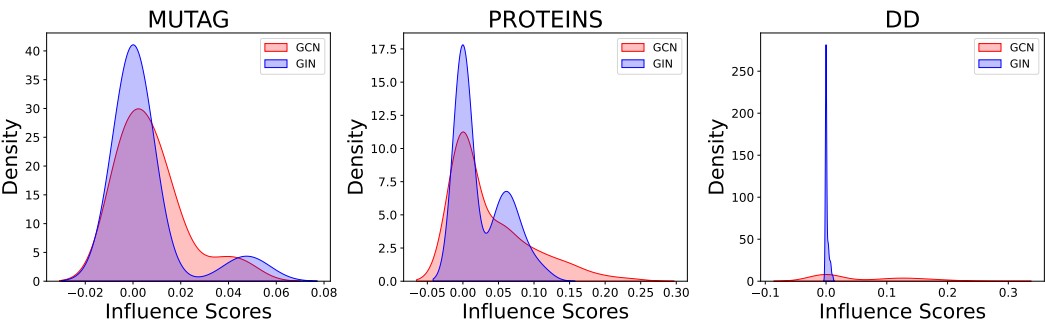

Figure 2: The density of the average influence scores of each augmented data on the test set.

## 4 EMPIRICAL EVALUATION

### 4.1 EXPERIMENTAL SETUP

**Datasets.** We evaluate our model on five widely used datasets from the GNN literature, specifically IMDB-BINARY, IMDB-MULTI, PROTEINS, MUTAG, and DD, all sourced from the TUD Benchmark (Morris et al., 2020). These datasets consist of either molecular or social graphs. Detailed statistics for each dataset are provided in Table 8 in Appendix F.

**Baselines.** We benchmark the performance of our approach against the state-of-the-art graph data augmentation strategies. In particular, we consider the DropNode (You et al., 2020), DropEdge (Rong et al., 2019), SubMix (Yoo et al., 2022), $\mathcal{G}$-Mixup (Han et al., 2022) and GeoMix (Zeng et al., 2024).

**Implementation Details.** We used the PyTorch Geometric (PyG) open-source library, licensed under MIT (Fey & Lenssen, 2019). The experiments were conducted on an RTX A6000 GPU. For the datasets from the TUD Benchmark, we used a size base split. We utilized two GNN architectures, GIN and GCN, both consisting of two layers with a hidden dimension of 32. The GNN was trained on graph classification tasks for 300 epochs with a learning rate of $10^{-2}$ using the Adam optimizer Kingma & Ba (2014). To model the graph representations of each class, we fit a GMM using the EM algorithm, running for 100 iterations or until the average lower bound gain dropped below $10^{-3}$. The number of Gaussians used in the GMM is provided in Table 9 of Appendix F. In 5 of Appendix D, we also present the performance of using the Variational Bayesian estimation (VB) instead of EM algorithm Tzikas et al. (2008). After generating new graph representations from each GMM, we fine-tuned the post-readout function for 100 epochs, maintaining the same learning rate of $10^{-2}$.

**Computation of Influence Scores.** Computing and inverting the Hessian matrix of the empirical risk is computationally expensive, with a complexity of $\mathcal{O}(N \times p^2 + p^3)$, where $p = |\theta|$ is the number of parameters in the GNN. To mitigate the cost of explicitly calculating the Hessian matrix, we employ implicit Hessian-vector products (iHVPs), following the approach outlined in Koh & Liang (2017).

### 4.2 EXPERIMENTAL RESULTS

**On the Generalization of GNN.** In Tables 2 and 1, we compare the test accuracy of our data augmentation strategy against baseline methods. Overall, our proposed approach consistently achieves the best or highly competitive performance for most of the datasets. Additionally, we observed that the results of the baseline methods vary depending on the GNN backbone, motivating further investigation using influence functions. As demonstrated in Theorem 3.3, the gradient, and more generally, the model architecture, significantly influence how augmented data impacts the model's performance on the test set.

**Robustness to Structure Corruption.** Besides generalization, we assess the robustness of our data augmentation strategy, following the methodology outlined by (Zeng et al., 2024). Specifically, we test the robustness of data augmentation strategies against graph structure corruption by randomly removing or adding 10% or 20% of the edges in the training set. By corrupting only the training graphs, we introduce a distributional shift between the training and testing datasets. This approach

Table 1: Classification accuracy ($\pm$ standard deviation) on different benchmark node classification datasets for the data augmentation baselines based on the GCN backbone. The higher the accuracy (in %) the better the model. Highlighted are the **first**, second best results.

| Model | IMDB-BINARY | IMDB-MULTI | MUTAG | PROTEINS | DD |
|---|---|---|---|---|---|
| No Aug. | 73.00 (4.94) | 47.73 (2.64) | 73.92 (5.09) | 69.99 (5.35) | 69.69 (2.89) |
| DropEdge | 71.70 (5.42) | 45.67 (2.46) | 73.39 (8.86) | 70.07 (3.86) | 69.35 (3.37) |
| DropNode | **74.00 (3.44)** | 43.80 (3.54) | 73.89 (8.53) | 69.81 (4.61) | 69.01 (3.95) |
| SubMix | 72.70 (5.59) | 46.00 (2.44) | 77.13 (9.69) | 67.57 (4.56) | 70.11 (4.48) |
| $\mathcal{G}$-Mixup | 72.10 (3.27) | 48.33 (3.06) | **88.77 (5.71)** | 65.68 (5.03) | 61.20 (3.88) |
| GeoMix | 69.69 (3.37) | 49.80 (4.71) | 74.39 (7.37) | 69.63 (5.37) | 68.50 (3.74) |
| GMM-GDA | 71.00 (4.40) | **49.82 (4.26)** | 76.05 (6.47) | **70.97 (5.07)** | **71.90 (2.81)** |

Table 2: Classification accuracy ($\pm$ standard deviation) on different benchmark node classification datasets for the data augmentation baselines based on the GIN backbone. The higher the accuracy (in %) the better the model. Highlighted are the **first**, second best results.

| Model | IMDB-BINARY | IMDB-MULTI | MUTAG | PROTEINS | DD |
|---|---|---|---|---|---|
| No Aug. | 70.30 (3.66) | 48.53 (4.05) | 83.42 (11.82) | 69.54 (3.61) | 68.00 (3.18) |
| DropEdge | 70.40 (4.03) | 46.80 (3.91) | 74.88 (9.62) | 68.27 (5.21) | 67.82 (4.46) |
| DropNode | 70.30 (3.49) | 45.20 (4.24) | 75.53 (7.89) | 65.40 (4.71) | **69.01 (3.95)** |
| SubMix | **72.50 (4.98)** | 48.13 (2.12) | 81.90 (9.21) | 70.44 (2.58) | 68.59 (5.04) |
| $\mathcal{G}$-Mixup | 70.70 (3.10) | 47.73 (4.95) | 87.77 (7.48) | 68.82 (3.48) | 63.91 (2.09) |
| GeoMix | 70.60 (4.61) | 47.20 (3.75) | 81.90 (7.55) | 69.80 (5.33) | 68.34 (5.30) |
| GMM-GDA | 71.70 (4.24) | **49.20 (2.06)** | **88.83 (5.02)** | **71.33 (5.04)** | 68.61 (4.62) |

allows us to evaluate GMM-GDA's ability to generalize well and predict the labels of test graphs, which can be considered OOD examples. The results of these experiments are presented in Table 3 for the IMDB-BINARY, IMDB-MULTI, PROTEINS, and DD datasets. As noted, our data augmentation strategy exhibits the best test accuracy in all cases and improves model robustness against structure corruption.

**Influence Functions.** In Figure 2, we show the density distribution of the average influence of augmented data sampled using GMM-GDA. For the MUTAG and PROTEINS datasets, we observe that GMM-GDA data augmentation has a positive impact on both GCN and GIN models. In contrast, for the DD dataset, GMM-GDA shows no effect on GIN, while it generates many augmented samples with positive values of the *influence scores* on GCN, thereby enhancing its performance. These findings are consistent with the empirical results presented in Tables 1 and 2.

**Configuration Models.** As part of an ablation study, we propose a simple yet effective graph augmentation strategy inspired by *Configuration Models* (Newman, 2013). As shown in Theorem 3.1, the objective is to control the term $\mathbb{E}_{\mathcal{G}\sim G}\mathbb{E}_{\lambda\sim\mathcal{P}}\left[\|\mathbf{h}_{\mathcal{G}^\lambda} - \mathbf{h}_{\mathcal{G}}\|\right]$, which can be achieved by regulating the distance between the original and the sampled graph within the input manifold, i.e., $\mathbb{E}_{\mathcal{G}\sim G}\mathbb{E}_{\lambda\sim\mathcal{P}}\left[\|\mathcal{G}^\lambda - \mathcal{G}\|\right]$. The approach involves generating a sampled version of each training graph while preserving its label by breaking a fraction $q$ of the existing edges (a percentage of the total number of edges $|\mathcal{E}_n|$) into *half-edges*, using a Bernoulli prior distribution $\mathcal{B}(r)$ with probability $r$. This process continues until all half-edges are connected. The strength of this method lies in its simplicity and in preserving the degree distribution, as the degree of each node and the total number of edges in the graph remain unchanged. If the distance norm in the input manifold is the $L_1$ distance between adjacency matrix, $|\mathcal{E}| \times r \times r$ is an upper bound of $\mathbb{E}_{\mathcal{G}\sim G}\mathbb{E}_{\lambda\sim\mathcal{P}}\left[\|\mathcal{G}^\lambda - \mathcal{G}\|\right]$, where $|\mathcal{E}|$ is the average of number of edges in training graphs. The results of this experiment are available in Appendix C. As noticed, the configuration model-based graph augmentation method performs competitively with the baselines and even outperforms them in certain cases. This underscores the importance of Theorem 3.1. When compared to our approach GMM-GDA, the latter gives better results across different datasets and GNN backbones. This difference is primarily due to the configuration model based approach being model-agnostic, whereas GMM-GDA leverages the model's weights and architecture, as explained in Section 3.4 and supported by Theorem 3.3.

Table 3: Robustness against structure corruption: We present the Classification accuracy ($\pm$ standard deviation). We highlighted the best data augmentation strategy **bold**. For this experiment, we use the GCN backbone.

| Noise Budget | 10% | | | | 20% | | | |
|---|---|---|---|---|---|---|---|---|
| Dataset | IMDB-BINARY | IMDB-MULTI | PROTEINS | DD | IMDB-BINARY | IMDB-MULTI | PROTEINS | DD |
| DropNode | 66.40 (5.51) | 44.46 (2.13) | 69.18 (4.87) | 65.79 (3.23) | 64.80 (5.01) | 43.06 (2.86) | 67.73 (6.43) | 64.35 (4.56) |
| DropEdge | 66.70 (5.10) | 43.80 (3.11) | 69.36 (5.90) | 68.42 (4.76) | 63.20 (6.30) | 41.80 (3.15) | 68.10 (5.05) | 67.06 (2.53) |
| SubMix | 69.30 (3.76) | 46.73 (2.67) | 69.80 (4.73) | 68.04 (7.64) | 63.70 (5.64) | 43.73 (3.60) | 69.09 (4.58) | 59.18 (6.29) |
| GeoMix | 72.20 (5.19) | 49.20 (4.31) | 70.25 (4.75) | 68.00 (3.64) | 70.90 (3.85) | 48.86 (5.18) | 68.36 (6.01) | 67.31 (3.91) |
| $\mathcal{G}$-Mixup | 68.30 (5.13) | 45.53 (4.12) | 61.71 (5.81) | 51.26 (8.76) | 63.20 (5.54) | 44.00 (4.63) | 46.63 (5.05) | 43.71 (7.12) |
| NoisyGNN | 70.50 (4.71) | 40.66 (3.12) | 69.45 (4.32) | 64.18 (5.71) | 63.50 (5.43) | 38.66 (4.12) | 69.99 (3.78) | 63.24 (5.02) |
| GMM-GDA | **72.80 (2.99)** | **49.36 (4.53)** | **70.61 (4.30)** | **68.68 (3.72)** | **73.10 (3.04)** | **49.53 (3.54)** | **70.32 (4.04)** | **69.01 (3.09)** |

## 5 CONCLUSION

In this paper, we introduced a novel approach for graph data augmentation that enhances both the generalization and robustness of GNNs. Our method uses Gaussian Mixture Models (GMMs) applied at the output level of the Readout function, an approach motivated by theoretical findings. Using the universal approximation property of GMMs, we can sample new graph representations to effectively control the upper bound of Rademacher Complexity, ensuring improved generalization of Graph Neural Networks (GNNs), as shown in Theorem 3.1. Through extensive experiments on widely used datasets, we demonstrated that our approach not only exhibits strong generalization ability but also maintains robustness against structural perturbations. An additional advantage of our method is its efficiency in terms of time complexity. Unlike baselines that generate augmented data for each individual or pair of training graphs, our approach fits the GMM to the entire training dataset at once, allowing for fast graph data augmentation without incurring significant additional backpropagation time.

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

## A   PROOF OF THEOREM 3.1

In this section, we provide a detailed proof of Theorem 3.1, aiming to derive a theoretical upper bound for both the generalization gap and the Rademacher complexity. **Theorem 3.1** Let $\ell$ be a classification loss function with $L_{Lip}$ as a Lipschitz constant such as $\ell(\cdot, \cdot) \in [0, 1]$. Then, with a probability at least $1 - \delta$ over the samples $\mathcal{D}_{train}$, we have,

$$\mathbb{E}_{\mathcal{G} \sim G} \left[ \ell(\mathcal{G}, \hat{\theta}_{aug}) \right] - \mathbb{E}_{\mathcal{G} \sim G} \left[ \ell(\mathcal{G}, \theta_\star) \right] \leq 2\mathcal{R}(\ell_{aug}) + 5\sqrt{\frac{2log(4/\delta)}{N}} + 2L_{Lip} \mathbb{E}_{\mathcal{G} \sim G} \mathbb{E}_{\lambda \sim \mathcal{P}} \left[ \left\| \mathcal{G}^\lambda - \mathcal{G} \right\| \right].$$

Moreover, we have,

$$\mathcal{R}(\ell_{aug}) \leq \mathcal{R}(\ell) + \max_{n \in \{1, \ldots, N\}} L_{Lip} \mathbb{E}_{\lambda \sim \mathcal{P}} \left[ \left\| \mathcal{G}_n^\lambda - \mathcal{G}_n \right\| \right].$$

*Proof.* We will decompose $\mathbb{E}_{\mathcal{G} \sim G} \left[ \ell(\mathcal{G}, \hat{\theta}_{aug}) \right] - \mathbb{E}_{\mathcal{G} \sim G} \left[ \ell(\mathcal{G}, \theta_\star) \right]$ into a finite sum of 5 terms as follows,

$$\mathbb{E}_{\mathcal{G} \sim G} \left[ \ell(\mathcal{G}, \hat{\theta}_{aug}) \right] - \mathbb{E}_{\mathcal{G} \sim G} \left[ \ell(\mathcal{G}, \theta_\star) \right] = u_1 + u_2 + u_3 + u_4 + u_5$$

where,

$$u_1 = \mathbb{E}_{\mathcal{G} \sim G} \left[ \ell(\mathcal{G}, \hat{\theta}_{aug}) \right] - \mathbb{E}_{\mathcal{G} \sim G} \left[ \mathbb{E}_{\lambda \sim \mathcal{P}} \left[ \ell(\mathcal{G}^\lambda, \hat{\theta}_{aug}) \right] \right],$$

$$u_2 = \mathbb{E}_{\mathcal{G} \sim G} \left[ \mathbb{E}_{\lambda \sim \mathcal{P}} \left[ \ell(\mathcal{G}^\lambda, \hat{\theta}_{aug}) \right] \right] - \frac{1}{N} \sum_{n=1}^{N} \mathbb{E}_{\lambda \sim \mathcal{P}} \left[ \ell(\mathcal{G}_n^\lambda, \hat{\theta}_{aug}) \right],$$

$$u_3 = \frac{1}{N} \sum_{n=1}^{N} \mathbb{E}_{\lambda \sim \mathcal{P}} \left[ \ell(\mathcal{G}_n^\lambda, \hat{\theta}_{aug}) \right] - \frac{1}{N} \sum_{n=1}^{N} \mathbb{E}_{\lambda \sim \mathcal{P}} \left[ \ell(\mathcal{G}_n^\lambda, \theta_\star) \right],$$

$$u_4 = \frac{1}{N} \sum_{n=1}^{N} \mathbb{E}_{\lambda \sim \mathcal{P}} \left[ \ell(\mathcal{G}_n^\lambda, \theta_\star) \right] - \mathbb{E}_{\mathcal{G} \sim G} \left[ \mathbb{E}_{\lambda \sim \mathcal{P}} \left[ \ell(\mathcal{G}^\lambda, \theta_\star) \right] \right],$$

$$u_5 = \mathbb{E}_{\mathcal{G} \sim G} \left[ \mathbb{E}_{\lambda \sim \mathcal{P}} \left[ \ell(\mathcal{G}^\lambda, \theta_\star) \right] \right] - \mathbb{E}_{\mathcal{G} \sim G} \left[ \ell(\mathcal{G}, \theta_\star) \right].$$

We upperbound each of the terms in the sum. We get,

$$u_1 + u_5 = \mathbb{E}_{\mathcal{G} \sim G} \left[ \ell(\mathcal{G}, \hat{\theta}_{aug}) \right] - \mathbb{E}_{\mathcal{G} \sim G} \left[ \mathbb{E}_{\lambda \sim \mathcal{P}} \left[ \ell(\mathcal{G}^\lambda, \hat{\theta}_{aug}) \right] \right] + \mathbb{E}_{\mathcal{G} \sim G} \left[ \mathbb{E}_{\lambda \sim \mathcal{P}} \left[ \ell(\mathcal{G}^\lambda, \theta_\star) \right] \right] - \mathbb{E}_{\mathcal{G} \sim G} \left[ \ell(\mathcal{G}, \theta_\star) \right]$$

$$\leq \left| \mathbb{E}_{\mathcal{G} \sim G} \left[ \ell(\mathcal{G}, \hat{\theta}_{aug}) \right] - \mathbb{E}_{\mathcal{G} \sim G} \left[ \mathbb{E}_{\lambda \sim \mathcal{P}} \left[ \ell(\mathcal{G}^\lambda, \hat{\theta}_{aug}) \right] \right] \right| + \left| \mathbb{E}_{\mathcal{G} \sim G} \left[ \mathbb{E}_{\lambda \sim \mathcal{P}} \left[ \ell(\mathcal{G}^\lambda, \theta_\star) \right] \right] - \mathbb{E}_{\mathcal{G} \sim G} \left[ \ell(\mathcal{G}, \theta_\star) \right] \right|$$

$$\leq 2 \sup_{\theta \in \Theta} \left| \mathbb{E}_{\mathcal{G} \sim G} \left[ \mathbb{E}_{\lambda \sim \mathcal{P}} \left[ \ell(\mathcal{G}^\lambda, \theta) \right] \right] - \mathbb{E}_{\mathcal{G} \sim G} \left[ \ell(\mathcal{G}, \theta) \right] \right|$$

$$\leq 2 \sup_{\theta \in \Theta} \left| \mathbb{E}_{\mathcal{G} \sim G} \left[ \mathbb{E}_{\lambda \sim \mathcal{P}} \left[ \ell(\mathcal{G}^\lambda, \theta) \right] - \ell(\mathcal{G}, \theta) \right] \right|$$

$$\leq 2 \sup_{\theta \in \Theta} \left| \mathbb{E}_{\mathcal{G} \sim G} \left[ \mathbb{E}_{\lambda \sim \mathcal{P}} \left[ \ell(\mathcal{G}^\lambda, \theta) - \ell(\mathcal{G}, \theta) \right] \right] \right|$$

$$\leq 2 L_{Lip} \sup_{\theta \in \Theta} \mathbb{E}_{\mathcal{G} \sim G} \mathbb{E}_{\lambda \sim \mathcal{P}} \left[ \left\| \mathcal{G}^\lambda - \mathcal{G} \right\| \right].$$

For the term $u_4$, we apply McDiarmid's inequality. Since the classification loss satisfy $\ell(\cdot) \in [0, 1]$, we get for $k \in \{0, \ldots, N\}$,

$\forall \{(\mathcal{G}_n, y_n)\}_{n=1}^{N}, \{(\mathcal{G}'_n, y'_n)\}_{n=1}^{N}, \theta$, such that $\forall n \neq k$,   $\mathcal{G}_n = \mathcal{G}'_n$ and $\mathcal{G}_k \neq \mathcal{G}'_k$:

$$\left| \frac{1}{N} \sum_{n=1}^{N} \mathbb{E}_{\lambda \sim \mathcal{P}} \left[ \ell(\mathcal{G}_n, \theta) \right] - \frac{1}{N} \sum_{n=1}^{N} \mathbb{E}_{\lambda \sim \mathcal{P}} \left[ \ell(\mathcal{G}'_n, \theta) \right] \right| = \frac{1}{N} \left| \sum_{n=1}^{N} \mathbb{E}_{\lambda \sim \mathcal{P}} \left[ \ell(\mathcal{G}_n, \theta) - \ell(\mathcal{G}'_n, \theta) \right] \right|$$

$$\leq \frac{1}{N} \left| \mathbb{E}_{\lambda \sim \mathcal{P}} \left[ \ell(\mathcal{G}_k, \theta) - \ell(\mathcal{G}'_k, \theta) \right] \right|$$

$$\leq 2/N.$$

The first equality is obtained by your claim that $\forall n \neq k, \quad \mathcal{G}_n = \mathcal{G'}_n$ and $\mathcal{G}_k \neq \mathcal{G'}_k$, the last inequality is obtained by the fact that $\ell(\cdot) \in [0, 1]$.

Thus,

$$\forall t > 0, \quad \mathbb{P}\left(u_4 \geq t\right) = \mathbb{P}\left(\frac{1}{N}\sum_{n=1}^{N}\mathbb{E}_{\lambda \sim \mathcal{P}}\left[\ell(\mathcal{G}_n^\lambda, \theta_\star)\right] - \mathbb{E}_{\mathcal{G}\sim G}\left[\mathbb{E}_{\lambda \sim \mathcal{P}}\left[\ell(\mathcal{G}^\lambda, \theta_\star)\right]\right] \geq t\right)$$

$$\leq exp\left(-\frac{2t^2}{\sum_{n=1}^{N} 4/N^2}\right)$$

$$= exp\left(-\frac{Nt^2}{2}\right).$$

Therefore, for $\delta \in\, ]0, 1]$, and for $t = \sqrt{2log(1/\delta)/(N)}$, i.e. $exp\left(-\frac{Nt^2}{2}\right) = \delta.$, we have,

$$\mathbb{P}\left(u_4 \geq \sqrt{2log(1/\delta)/(N)}\right) \leq \delta.$$

Therefore,

$$\mathbb{P}\left(u_4 < \sqrt{\frac{2log(1/\delta)}{N}}\right) = 1 - \mathbb{P}\left(u_4 \geq \sqrt{\frac{2log(1/\delta)}{N}}\right) \geq 1 - \delta.$$

Thus, with a probability of at least $1 - \delta$,

$$u_4 \leq \sqrt{\frac{2log(1/\delta)}{N}} < \sqrt{\frac{2log(4/\delta)}{N}}.$$

Moreover, Rademacher complexity holds for $u_2$,

$$u_2 = \mathbb{E}_{\mathcal{G}\sim G}\left[\mathbb{E}_{\lambda \sim \mathcal{P}}\left[\ell(\mathcal{G}^\lambda, \hat{\theta}_{aug})\right]\right] - \frac{1}{N}\sum_{n=1}^{N}\mathbb{E}_{\lambda \sim \mathcal{P}}\left[\ell(\mathcal{G}_n^\lambda, \hat{\theta}_{aug})\right] \leq 2\mathcal{R}(\ell_{aug}) + 4\sqrt{\frac{2log(4/\delta)}{N}}.$$

The above inequality tells us that the true risk $\mathbb{E}_{\mathcal{G}\sim G}\left[\mathbb{E}_{\lambda \sim \mathcal{P}}\left[\ell(\mathcal{G}^\lambda, \hat{\theta}_{aug})\right]\right]$ is bounded by the empirical risk $\frac{1}{N}\sum_{n=1}^{N}\mathbb{E}_{\lambda \sim \mathcal{P}}\left[\ell(\mathcal{G}_n^\lambda, \hat{\theta}_{aug})\right]$ plus a term depending on the Rademacher complexity of the augmented hypothesis class and an additional term that decreases with the size of the sample $N$.

Additionally, since $\hat{\theta}_{aug}$ is the optimal parameter for the loss $\frac{1}{N}\sum_{n=1}^{N}\mathbb{E}_{\lambda \sim \mathcal{P}}\left[\ell(\mathcal{G}_n^\lambda, \theta)\right]$, thus,

$$u_3 \leq 0$$

By summing all the inequalities, we conclude that,

$$\mathbb{E}_{\mathcal{G}\sim G}\left[\ell(\mathcal{G}, \hat{\theta}_{aug})\right] - \mathbb{E}_{\mathcal{G}\sim G}\left[\ell(\mathcal{G}, \theta_\star)\right] < 2\mathcal{R}(\ell_{aug}) + 5\sqrt{\frac{2log(4/\delta)}{N}} + 2L_{Lip}\mathbb{E}_{\mathcal{G}\sim G}\mathbb{E}_{\lambda \sim \mathcal{P}}\left[\left\|\mathcal{G}_n^\lambda - \mathcal{G}_n\right\|\right].$$

Part 2 of the proof.

$$\mathcal{R}(\ell_{aug}) - \mathcal{R}(\ell) = \mathbb{E}_{\epsilon_n \sim P_\epsilon} \left[ \sup_{\theta \in \Theta} \left| \frac{1}{N} \sum_{n=1}^N \epsilon_n \ell_{aug}(\mathcal{G}_n, \theta) \right| - \sup_{\theta \in \Theta} \left| \frac{1}{N} \sum_{n=1}^N \epsilon_n \ell(\mathcal{G}_n, \theta) \right| \right]$$

$$\leq \mathbb{E}_{\epsilon_n \sim P_\epsilon} \left[ \sup_{\theta \in \Theta} \left| \frac{1}{N} \sum_{n=1}^N \epsilon_n \ell_{aug}(\mathcal{G}_n, \theta) - \frac{1}{N} \sum_{n=1}^N \epsilon_n \ell(\mathcal{G}_n, \theta) \right| \right]$$

$$= \mathbb{E}_{\epsilon_n \sim P_\epsilon} \left[ \sup_{\theta \in \Theta} \left| \frac{1}{N} \sum_{n=1}^N \epsilon_n \left( \ell_{aug}(\mathcal{G}_n, \theta) - \ell(\mathcal{G}_n, \theta) \right) \right| \right]$$

$$\leq \mathbb{E}_{\epsilon_n \sim P_\epsilon} \left[ \sup_{\theta \in \Theta} \frac{1}{N} \sum_{n=1}^N \left| \epsilon_n \left( \ell_{aug}(\mathcal{G}_n, \theta) - \ell(\mathcal{G}_n, \theta) \right) \right| \right]$$

$$\leq \sup_{\theta \in \Theta} \frac{1}{N} \sum_{n=1}^N \left| \ell_{aug}(\mathcal{G}_n, \theta) - \ell(\mathcal{G}_n, \theta) \right|$$

$$= \sup_{\theta \in \Theta} \frac{1}{N} \sum_{n=1}^N \left| \mathbb{E}_{\lambda \sim \mathcal{P}} \left[ \ell(\mathcal{G}_n^\lambda, \theta) - \ell(\mathcal{G}_n, \theta) \right] \right|$$

$$\leq \max_{n \in \{1, \ldots, N\}} L_{Lip} \mathbb{E}_{\lambda \sim \mathcal{P}} \left[ \left\| \mathcal{G}_n^\lambda - \mathcal{G}_n \right\| \right].$$

$\square$

## B    PROOF OF THEOREM 3.3

In this section, we present the detailed proof of Theorem 3.3, which allows us to e perform an in-depth theoretical analysis of our augmentation strategy through the lens of influence functions.

**Theorem 3.3** Given a test graph $\mathcal{G}_k$ from the test set, let $\hat{\theta} = \arg \min_\theta \mathcal{L}$ be the GNN parameters that minimize the objective function in Equation 3. The impact of upweighting the objective function $\mathcal{L}$ to $\mathcal{L}_{i,j}^{\text{aug}} = \mathcal{L} + \epsilon_{n,m} \ell(\mathcal{G}_n^{\lambda_{n,m}}, \theta)$, where $\mathcal{G}_n^{\lambda_{n,m}}$ is an augmented graph candidate of the training graph $\mathcal{G}_n$ and $\epsilon_{n,m}$ is a sufficiently small perturbation parameter, on the model performance on the test graph $\mathcal{G}_k^{test}$ is given by

$$\frac{d\ell(\mathcal{G}_k^{test}, \hat{\theta}_{\epsilon,m})}{d\epsilon_{n,m}} = -\nabla_\theta \ell(\mathcal{G}_k^{test}, \hat{\theta}) H_{\hat{\theta}}^{-1} \nabla_\theta \ell(\mathcal{G}_n^{\lambda_{n,m}}, \hat{\theta}),$$

where $\hat{\theta}_{\epsilon_{n,m}} = \arg \min_\theta \mathcal{L}_{n,m}^{\text{aug}}$ denotes the parameters that minimize the upweighted objective function $\mathcal{L}_{n,m}^{\text{aug}}$ and $H_{\hat{\theta}} = \nabla_\theta^2 \mathcal{L}(\hat{\theta})$ is the Hessian Matrix of the loss w.r.t the model parameters.

*Proof.* Lets $\mathcal{G}_n^{\lambda_{n,m}}$ be an augmented graph candidate of the training graph $\mathcal{G}_n$ and $\epsilon_{n,m}$ is a sufficiently small perturbation parameter. The parameters $\hat{\theta}$ and $\hat{\theta}$ and $\hat{\theta}_{\epsilon_{n,m}}$ the parameters that minimize the empirical risk on the train set, i.e.,

$$\hat{\theta} = \arg \min_\theta \mathcal{L}.$$

$$\hat{\theta}_{\epsilon_{n,m}} = \arg \min_\theta \mathcal{L}_{n,m}^{\text{aug}} = \arg \min_\theta \mathcal{L} + \epsilon_{n,m} \ell(\mathcal{G}_n^{\lambda_{n,m}}, \theta).$$

Therefore, we examine its firstorder optimality conditions,

$$0 = \nabla_{\hat{\theta}} \mathcal{L} \tag{6}$$

$$0 = \nabla_{\hat{\theta}_{\epsilon_{n,m}}} \left( \mathcal{L} + \epsilon_{n,m} \ell(\mathcal{G}_n^{\lambda_{n,m}}, \theta) \right). \tag{7}$$

$\square$

Using Taylor Expansion, we now develop the Equation 7. We have $\lim_{\epsilon_{n,m} \to 0} \hat{\theta}_{\epsilon_{n,m}} = \hat{\theta}$, thus,

$$0 \simeq \left[\nabla_{\hat{\theta}}\mathcal{L}(\hat{\theta}) + \epsilon_{n,m}\nabla_{\hat{\theta}}\ell(\mathcal{G}_n^{\lambda_{n,m}}, \hat{\theta})\right] + \left[\nabla_{\hat{\theta}}^2\mathcal{L}(\hat{\theta}) + \epsilon_{n,m}\nabla_{\hat{\theta}}^2\ell(\mathcal{G}_n^{\lambda_{n,m}}, \hat{\theta})\right]\left(\hat{\theta}_{\epsilon_{n,m}} - \hat{\theta}\right).$$

Therefore,

$$\hat{\theta}_{\epsilon_{n,m}} - \hat{\theta} = -\left[\nabla_{\hat{\theta}}^2\mathcal{L}(\hat{\theta}) + \epsilon_{n,m}\nabla_{\hat{\theta}}^2\ell(\mathcal{G}_n^{\lambda_{n,m}}, \hat{\theta})\right]^{-1}\left[\nabla_{\hat{\theta}}\mathcal{L}(\hat{\theta}) + \epsilon_{n,m}\nabla_{\hat{\theta}}\ell(\mathcal{G}_n^{\lambda_{n,m}}, \hat{\theta})\right].$$

Dropping the $\circ(\epsilon_{n,m})$ terms, and using the Equation 6, i.e. $\nabla_{\hat{\theta}}\mathcal{L} = 0$, we conclude that,

$$\frac{\hat{\theta}_{\epsilon_{n,m}} - \hat{\theta}}{\epsilon_{n,m}} = -\left[\nabla_{\hat{\theta}}^2\mathcal{L}(\hat{\theta})\right]^{-1}\nabla_{\hat{\theta}}\ell(\mathcal{G}_n^{\lambda_{n,m}}, \hat{\theta}).$$

Therefore,

$$\frac{d\hat{\theta}_{\epsilon_{n,m}}}{d\epsilon_{n,m}} \simeq \frac{\hat{\theta}_{\epsilon_{n,m}} - \hat{\theta}}{\epsilon_{n,m}} = -\left[\nabla_{\hat{\theta}}^2\mathcal{L}(\hat{\theta})\right]^{-1}\nabla_{\hat{\theta}}\ell(\mathcal{G}_n^{\lambda_{n,m}}, \hat{\theta}).$$

$$\frac{d\ell(\mathcal{G}_k^{test}, \hat{\theta}_{\epsilon_{n,m}})}{d\epsilon_{n,m}} = \frac{d\ell(\mathcal{G}_k^{test}, \hat{\theta}_{\epsilon_{n,m}})}{d\hat{\theta}_{\epsilon_{n,m}}}\frac{d\hat{\theta}_{\epsilon_{n,m}}}{d\epsilon_{n,m}}.$$

## C  CONFIGURATION MODELS

In this section, we present a novel adaptation of Configuration Models as a graph data augmentation technique for GNN. Configuration Models Newman (2013) enable the generation of randomized graphs that maintain the original degree distribution. We can, therefore, leverage this strategy to improve the generalization of GNNs. Below, we present the steps involved in our approach to using Configuration Models for Graph Data Augmentation:

1. **Extract Edges:** For each training graph $\mathcal{G}_n$, we first extract the complete set of edges $\mathcal{E}_n$.

2. **Stub Creation:** Using a Bernoulli distribution with parameter $p \in [0, 1]$, we randomly select a subset of candidate edges and *break* them to create *stubs* (half-edges).

3. **Stub Pairing:** We then randomly pair these stubs to form new edges, creating a randomized graph structure with the same degree distribution.

Table 4 shows the performance of this approach on the two GNN backbone GCN and GIN.

Table 4: Classification accuracy ($\pm$ standard deviation) on different benchmark node classification datasets for the data augmentation baselines based on the GIN backbone. The higher the accuracy (in %) the better the model.

| Model | IMDB-BINARY | IMDB-MULTI | MUTAG | PROTEINS | DD |
|---|---|---|---|---|---|
| Config Models w/ GCN | 71.70 (3.16) | 48.40 (3.88) | 74.97 (6.77) | 70.08 (4.93) | 69.01 (3.44) |
| Config Models w/ GIN | 71.70 (4.24) | 49.00 (3.44) | 81.43 (10.05) | 68.34 (5.30) | 71.61 (5.96) |

## D  ABLATION STUDY

To provide additional comparison and motivate the use of GMMs with the EM algorithm, we expanded our evaluation to include additional methods for modeling the distribution of the graph representations. Specifically, the comparison includes:

- **GMM w/ Variational Bayesian Inference (VBI):** We specifically compared the Expectation-Maximization (EM) algorithm, discussed in the main paper, with the Variational Bayesian (VB) estimation technique for parameter estimation of each Gaussian

Mixture Model (GMM) (Tzikas et al., 2008) for both the GCN and GIN models. The objective of including this baseline is to explore alternative approaches for fitting GMMs to the graph representations.

- **Kernel Density Estimation (KDE) :** KDE is a Neighbor-Based Method and a non-parametric approach to estimating the probability density (Härdle et al., 2004). KDE estimates the probability density function by placing a kernel function (e.g., Gaussian) at each data point. The sum of these kernels approximates the underlying distribution. Sampling can be done using techniques like Metropolis-Hastings. The purpose of using KDE as a baseline is to evaluate alternative distributions different from the Gaussian Mixture Model (GMM).

- **Copula-Based Methods:** We model the dependence structure between variables using copulas, while marginal distributions are modeled separately. We sample from marginal distributions and then transform them using the copula (Nelsen, 2006).

- **Generative Adversarial Network (GAN):** GANs are powerful generative models that learn to approximate the data distribution through an adversarial process between two neural networks. To evaluate the performance of deep learning-based generative approaches for modeling graph representations, we included tGAN, a GAN architecture specifically designed for tabular data (Yang et al., 2012). We particularly train tGAN on the graph representations and then sample new graph representations from the generator.

Table 5: Ablation Study on the density estimation scheme for learned GCN representations.

| Model | IMDB-BINARY | IMDB-MULTI | MUTAG | PROTEINS | DD |
|---|---|---|---|---|---|
| GMM w/ EM | **71.00 (4.40)** | **49.82 (4.26)** | **76.05 (6.47)** | **70.97 (5.07)** | **71.90 (2.81)** |
| GMM w/ VBI | **71.00 (4.21)** | 49.53 (4.26) | **76.05 (6.47)** | **70.97 (4.52)** | 71.64 (2.90) |
| KDE | 55.90 (10.29) | 39.53 (2.87) | 66.64 (6.79) | 59.56 (2.62) | 58.66 (3.97) |
| Copula | 69.80 (4.04) | 47.13 (3.45) | 74.44 (6.26) | 65.04 (3.37) | 65.70 (3.04) |
| GAN | 70.60 (3.41) | 48.80 (5.51) | 75.52 (4.96) | 69.98 (5.46) | 66.26 (3.72) |

Table 6: Ablation Study on the density estimation scheme for learned GIN representations.

| Model | IMDB-BINARY | IMDB-MULTI | MUTAG | PROTEINS | DD |
|---|---|---|---|---|---|
| GMM w/ EM | **71.70 (4.24)** | **49.20 (2.06)** | **88.83 (5.02)** | **71.33 (5.04)** | **68.61 (4.62)** |
| GMM w/ VBI | 71.40 (2.65) | 47.80 (2.22) | 88.30 (5.19) | 70.25 (4.65) | 67.82 (4.96) |
| KDE | 69.10 (3.93) | 41.46 (3.02) | 77.60 (6.83) | 60.37 (3.04) | 67.48 (6.18) |
| Copula | 70.60 (2.61) | 47.60 (2.29) | 88.30 (5.19) | 70.16 (4.55) | 67.91 (4.90) |
| GAN | 70.50 (3.80) | 48.40 (1.71) | **88.83 (5.02)** | **71.33 (5.55)** | 67.74 (4.82) |

We compare these approaches for both the GCN and GIN models in Tables 5 and 6, respectively. As noticed, GMM with EM consistently outperforms the alternative methods across most datasets in terms of accuracy. The VBI method, an alternative approach for estimating GMM parameters, yields comparable performance to the EM algorithm. This consistency across datasets highlights the effectiveness and robustness of GMMs in capturing the underlying data distribution.

In certain cases, particularly with the GIN model, we observed competitive performance from the GAN approach, which, unlike GMM, requires additional training. Hence, GMMs provide a more straightforward and efficient solution.

## E    TRAINING AND AUGMENTATION TIME

We compare the data augmentation times of our approach and the baselines in Table 7. In addition to outperforming the baselines on most datasets, our approach offers an advantage in terms of time complexity. The training time of baseline models varies depending on the augmentation strategy used, specifically, whether it involves pairs or individual graphs. Even in cases where a graph augmentation has a low computational cost for some baselines, training can still be time-consuming as multiple augmented graphs are required to achieve satisfactory test accuracy. For instance, methods like

DropEdge, DropNode, and SubMix, while computationally simple, require generating multiple augmented samples at each epoch, thereby increasing the overall training time. In contrast, GMM-GDA introduces a more efficient approach by generating only one augmented graph per training instance, which is reused across all epochs. This design ensures a balance between computational efficiency and augmentation effectiveness, reducing the overall training burden while maintaining strong performance. The only baseline that is more time-efficient than our approach is GeoMix; however, our method consistently outperforms GeoMix across all settings, as shown in Tables 1 and 2.

Table 7: Mean training and augmentation time in seconds of our model in comparison to the other benchmarks.

|   | Attack | Model | IMDB-BINARY | MUTAG | DD |
|---|---|---|---|---|---|
| ① | Aug. Time | Vanilla | - | - | - |
|   |   | DropEdge | 0.02 | 0.01 | 0.01 |
|   |   | DropNode | 0.01 | 0.02 | 0.01 |
|   |   | SubMix | 1.27 | 0.23 | |
|   |   | $\mathcal{G}$-Mixup | 0.74 | 0.11 | 4.26 |
|   |   | GeoMix | 2,344.12 | 73.52 | 1,005.35 |
|   |   | GMM-GDA | 2.87 | 0.51 | 3.25 |
| ② | Train. Time | Vanilla | 765.96 | 99.32 | 428.10 |
|   |   | DropEdge | 892.14 | 596.82 | 3,037.30 |
|   |   | DropNode | 884.71 | 803.63 | 3,325 |
|   |   | SubMix | 1,711.01 | 1,487.03 | |
|   |   | $\mathcal{G}$-Mixup | 148.71 | 28.14 | 177.55 |
|   |   | GeoMix | 89.01 | 101.82 | 123.41 |
|   |   | GMM-GDA | 774.47 | 101.56 | 438.39 |

## F DATASETS AND IMPLEMENTATION DETAILS

### F.1 GRAPH CLASSIFICATION

Characteristics and information about the datasets utilized for the graph classification task are presented in Table 8. As outlined in the main paper, we conduct experiments on IMDB-BINARY, IMDB-MULTI, PROTEINS, MUTAG, and DD, all sourced from the TUD Benchmark Ivanov et al. (2019). These datasets consist of either molecular or social graphs.

Table 8: Statistics of the graph classification datasets used in our experiments.

| Dataset | #Graphs | Avg. Nodes | Avg. Edges | #Classes |
|---|---|---|---|---|
| IMDB-BINARY | 1,000 | 19.77 | 96.53 | 2 |
| IMDB-MULTI | 1,500 | 13.00 | 65.94 | 3 |
| MUTAG | 188 | 17.93 | 19.79 | 2 |
| PROTEINS | 1,113 | 39.06 | 72.82 | 2 |
| DD | 1,178 | 284.32 | 715.66 | 2 |

### F.2 IMPLEMENTATION DETAILS

For all the used models, the same number of layers, hyperparameters, and activation functions were used. The models were trained using the cross-entropy loss function with the Adam optimizer, the number of epochs and learning rate were kept similar for the different approaches across all experiments. In Table 9, we present the optimal number of Gaussian distributions in the GMM for each dataset and GNN backbone

Table 9: The optimal number of Gaussian distributions in the GMM for each pair of dataset and GNN backbone.

| Dataset | IMDB-BINARY | IMDB-MULTI | MUTAG | PROTEINS | DD |
|---------|-------------|------------|-------|----------|-----|
| GCN | 40 | 50 | 10 | 10 | 2 |
| GIN | 50 | 5 | 2 | 2 | 50 |

# G  GRAPH DISTANCE METRICS

Let us consider the graph space $(\mathbb{G}, \|\cdot\|_\mathbb{G})$ and the feature space $(\mathbb{X}, \|\cdot\|_\mathbb{X})$, where $\|\cdot\|_\mathbb{G}$ and $\|\cdot\|_\mathbb{X}$ denote the norms applied to the graph structure and features, respectively. When considering only structural changes, with fixed node features, the distance between two graphs $\mathcal{G}^\lambda, \mathcal{G}$ is defined as

$$\left\|\mathcal{G}^\lambda - \mathcal{G}\right\| = \|A - A^\lambda\|_\mathbb{G}, \tag{8}$$

where $A^\lambda, A$ are respectively the adjacency matrix of $\mathcal{G}^\lambda, \mathcal{G}$, and the norm $\|\cdot\|_\mathbb{G}$ can be for example the Frobenius or spectral norm. If both structural and feature changes are considered, the distance extends to:

$$\left\|\mathcal{G}^\lambda - \mathcal{G}\right\| = \alpha\|A - A^\lambda\|_\mathbb{G} + \beta\|X - X^\lambda\|_\mathbb{X}, \tag{9}$$

where $X^\lambda, X$ are the node feature matrices of $\mathcal{G}^\lambda, \mathcal{G}$ respectively, and $\alpha, \beta$ are hyperparameters controlling the contribution of structural and feature differences.

In most baselines graph augmentation techniques, such as for instance $\mathcal{G}$-Mixup, SubMix, and DropNode, the alignment between nodes in the original graph $\mathcal{G}$ and the augmented graph $\mathcal{G}^\lambda$ is known. However, in cases where the node alignment is unknown, we must take into account node permutations. The distance between the two graphs is then defined as

$$\left\|\mathcal{G}^\lambda - \mathcal{G}\right\| = \min_{P \in \Pi} \left(\alpha\|A - PA^\lambda P^T\|_\mathbb{G} + \beta\|X - PX^\lambda\|_\mathbb{X}\right), \tag{10}$$

where $\Pi$ is the set of permutation matrices. The matrix $P$ corresponds to a permutation matrix used to order nodes from different graphs. By using Optimal Transport, we find the minimum distance over the set of permutation matrices, which corresponds to the optimal matching between nodes in the two graphs. This formulation represents the general case of graph distance, which has been used in the literature (Abbahaddou et al., 2024).

