# OpenReview forum: "Gaussian Mixture Models Based Augmentation Enhances GNN Generalization"
_ICLR.cc/2025/Conference — Submitted to ICLR 2025_

### Official Review · Reviewer_qdD6 · 2024-10-27

**Soundness:** 3
**Presentation:** 3
**Contribution:** 2
**Rating:** 6
**Confidence:** 4

**Summary:**

The authors propose a novel graph data augmentation based approach to tackle the graph OOD problem. To be specific, they first train a GNN model using the training data. Then, graphs within each class in the training data are fed to the GNN model and the output of the readout layer are treated as the embeddings of this class. After that, the authors propose to fit a Gaussian Mixture Model (GMM) on the embeddings for each class using the classical EM algorithm. Finally, the augmented embeddings are generated by sampling from the GMMs for each class, which are combined with the embeddings of training data and used for fine tuning the post-Readout function. The proposed framework enjoy a high computation efficiency since the post-Readout function contains only a linear layer and the (time) complexity of fitting GMMs is linear. The authors also provide some theoretic analysis. First, they analyze the excess risk of the graph augmentation approach and the result shows that minimizing the expected distance between original graphs and augmented ones could reduce the excess risk. Second, they use influence functions to quantify the affect of augmented data on model's performance on testing data. Experimental results show that the their proposed method has competitive performance against baselines and has a significant benefit on advantages in robustness against structure corruption and time complexity.

**Strengths:**

The proposed technique is reasonable and efficient. The experimental results provided in the paper are sufficient to support the effectiveness of this technique. Considering the balance between effectiveness and computational efficiency of this technique, it has a potential application on real-world scenarios. Besides, the paper is well-written and easy to follow.

**Weaknesses:**

Although using a generative models to learn the graph representation distribution is reasonable, the motivation of adopting GMMs is still unclear. GMM is universal approximator of densities could be one of the reasons, yet how many component need to achieve a small approximation error is unknown. Besides, the theoretic results provided in this paper do not seem to explain why this method that generating augmentations in representation space is superior or comparable to previous methods that generating augmentations in data space, which could be a promising direction to be explored.

**Questions:**

Q1: The number of Gaussian distributions $K$ in GMMs is an important hyperparameter and has impact on the performance of GMM. How do you properly choose this hyperparameter in practices? Please describe the tuning process of $K$ or provide hyperparameter sensitivity analysis to show how performance varies with different values of $K$.

Q2: In line 349, the subscripts $i,j$ of the notation $\\mathcal{L}^{aug}_{i,j}$ are ambiguous. I think the correct one should be $\\mathcal{L}^{aug} _ {n,m}$.

Q3: The proof between line 703 to line 715 seems confusing. The first equality only holds when the right hand side also includes a expectation w.r.t. $\lambda_{n,m}$. Indeed, I think the proof should be proceeded as
\begin{equation}
\left\Vert \\frac{1}{N} \\sum_{n=1}^N \\mathbb{E}_{\\lambda \sim \\mathcal{P}} [\ell(\\mathcal{G}_n,\theta) - \ell(\\mathcal{G}'_n,\theta) ] \right\Vert =  \\left\\Vert \\frac{1}{N}  \\mathbb{E} _ {\\lambda \\sim \\mathcal{P}} [\ell(\\mathcal{G} _ k,\theta) - \ell(\\mathcal{G}' _ k,\theta)] \\right\\Vert \leq \\frac{1}{N}  \\mathbb{E} _ {\\lambda \\sim \\mathcal{P}} [\Vert \ell(\\mathcal{G}_k,\theta) - \ell(\\mathcal{G}'_k,\theta) \Vert ] \leq \frac{1}{N}.
\end{equation}
The first equality is obtained by your claim that $\\mathcal{G} _ k = \\mathcal{G}' _ k$ for $k = 1,\ldots, N$ and $k \neq n$. The last inequality is obtained by $\ell(\cdot) \in [0,1]$.
Please clarify this issue or correct this part of proof following the above steps.

Q4: In line 754, you claim that $\\hat{\theta} _ {aug}$ is the optimal parameter of the loss $\frac{1}{N} \\sum_{n=1}^N \\mathbb{E} _ {\\lambda \sim \\mathcal{P}}  [\ell(\\mathcal{G}^{\\lambda} _ n,\theta)]$, which is different from the definition of $\\hat{\theta}_{aug}$ in line 195 where $\\hat{\theta} _ {aug} = \\mathop{\rm argmin} _ {\theta} \\frac{1}{NM} \\sum _ {n=1}^N \\sum _ {m=1}^M \ell(\\mathcal{G} _ n^{\\lambda _ {n,m}}, \theta)$. This could make the inequality $v_3 \leq 0$ do not hold. Please clarify and check the definition of $\\hat{\theta} _ {aug}$. If the above issue do exist, you should consider revising your proof accordingly.

Q5: In line 221-223, you claim that minimizing the term $\\mathbb{E} _ {\\mathcal{G} \sim G} \\mathbb{E} _ {\\lambda \sim \\mathcal{P}} [\Vert \\mathcal{G}^{\\lambda} - \\mathcal{G} \Vert ]$ can guarantee with a high probability to decrease both the Rademacher complexity and the generalization risk. And you also show that the Rademacher complexity term $\\mathcal{R}(\ell _ {aug})$ is upper bounded by $\\mathop{\rm max} _ {n=1,\ldots,N} \\mathbb{E} _ {\\lambda \sim \\mathcal{P}} [ \Vert \\mathcal{G}^{\\lambda} _ n - \\mathcal{G} _ n \Vert ]$, which is a empirical estimation of $\\mathbb{E} _ {\\mathcal{G} \sim G} \\mathbb{E} _ {\\lambda \sim \\mathcal{P}} [\Vert \\mathcal{G}^{\\lambda} - \\mathcal{G} \Vert ]$ w.r.t. $\\mathcal{G}$. Therefore, minimizing the term $\\mathbb{E} _ {\\mathcal{G} \sim G} \\mathbb{E} _ {\\lambda \sim \\mathcal{P}} [\Vert \\mathcal{G}^{\\lambda} - \\mathcal{G} \Vert ]$ may not guarantee to decrease the Rademacher complexity term $\\mathcal{R}(\ell _ {aug})$. Please clarify this issue or modify your claim in line 221-223.

---

> ### Author Response · Authors · 2024-11-22
> **Response to Reviewer qdD6**
>
> *We thank Reviewer  qdD6 for the feedback. In what follows, we address the raised questions and weaknesses point-by-point.*
>
>
>
> ***[Weaknesses] Motivation behind GMM-GDA***
>
> Indeed, other generative techniques, such as generative models, could also be used to fit the embeddings of the training data. However, we chose to use GMMs for several important reasons, particularly the efficiency and effectiveness of GMMs in this context.
>
> GMMs are universal approximators, meaning they can effectively approximate any distribution, including the distribution of the graph embeddings. This property ensures that the augmented data is drawn from a distribution that closely aligns with the original data distribution, as shown in Theorem 3.1. While other generative methods could be used to fit the embeddings, GMMs have the advantage of being relatively simple and efficient in terms of computation (we can fit a GMM and generate new samples in very few seconds). Unlike more complex methods such as generative models, GMMs can achieve high-quality approximations with minimal computational overhead.
>
>
> ***[Question 1] The number of Gaussian distribution***
>
> Theoretically, increasing $K$ allows the GMM to better approximate the true distribution of graph embeddings, as a higher number of components provides more flexibility in capturing complex distributions.  We experimented with $K$ values in the range of 2 to 50, with the maximum $K=50$ being sufficient to balance model complexity and computational feasibility. In some cases, even a small number of components (e.g., $K= 2$ or $K=5$) was sufficient to achieve competitive performance. Below, We include a hyperparameter sensitivity analysis on the GIN backbone and the dataset IMDB--BINARY and we noticed a consistency in the results.
>
> |     | K=10        | K=20        | K=30        | K=40        | K=50        |
> |-----|-------------|-------------|-------------|-------------|-------------|
> | GIN | 71.1 (2.70) | 71.3 (2.28) | 71.3 (2.57) | 71.6 (2.72) | 71.7 (2.64) |
>
>
>
> ***[Question 2] Typo***
>
> We are grateful to the reviewer for spotting the typo. We made the necessary adaptations in the submitted paper.
>
> ***[Question 3] Proof of Theorem 3.1***
>
> We thank the reviewer for pointing the error in the proof. We updated the proof in the paper with the necessary changes.
>
> ***[Question 4] Clarification on the definition of $\hat{\theta}_{aug}$***
>
> As defined in line 754 (in the Appendix) and in the mathematical formalism of graph data augmentation (in Section 3.1), the weights $\hat{\theta}_{aug}$ correspond to
>
> $ \text{argmin} \frac{1}{N} \sum_{n=1}^{N} \mathbb{E}_{\lambda  \sim \mathcal{P} }\left [ \ell(\mathcal{G}_n^\lambda,\theta) \right ],$
>
> and its empirically approximated by  $\text{argmin} \frac{1}{N\times M} \sum_{n=1}^N \sum_{m=1}^M  \ell(\mathcal{G}_n^{\lambda} {}^{n,m}, \theta).$
>
> The proof for this formulation still holds. We have provided a clearer definition of $\hat{\theta}_{aug}$ in Section 3 of the manuscript.
>
> ***[Question 5] The Upperbound in Theorem 3.1***
>
> It is true that if the maximum of the expectations $ \max_{n \in \{1, \ldots,N\}} \mathbb{E}_{\lambda \sim \mathcal{P}} \left [ \left \| \mathcal{G}_n^\lambda -\mathcal{G}_n \right \| \right ], \text{ is non-zero, the Rademacher complexity of},$
>
>  $\ell_{aug}$ may not necessarily be smaller than the Rademacher complexity of $\ell$.  However, by minimizing the term $ \mathbb{E}_{\lambda \sim \mathcal{P}} \left [ \left \| \mathcal{G}_n^\lambda -\mathcal{G}_n \right \| \right ],$
>
> we are more likely to reduce the additional Rademacher complexity, thus increasing the chances of achieving a lower overall Rademacher complexity. Since we use Gaussian Mixture Models (GMMs) for augmenting the graph data, which are universal approximators, we ensure that the expectations $ \max_{n \in \{1, \ldots,N\}} \mathbb{E}_{\lambda \sim \mathcal{P}} \left [ \left \| \mathcal{G}_n^\lambda -\mathcal{G}_n \right \| \right ]$ approach zero.

---

> > ### Comment · Reviewer_qdD6 · 2024-11-27
> > **Reply to the Authors**
> >
> > Thanks the authors for their detailed responses, which have addressed mostly my concerns on the correctness of the theoretic results in this work. I have increased my score to 6.

---

> > > ### Author Response · Authors · 2024-12-01
> > >
> > > Thank you for acknowledging our clarification and for raising your score. We greatly appreciate your constructive feedback.

---

### Official Review · Reviewer_YuEk · 2024-11-04

**Soundness:** 3
**Presentation:** 3
**Contribution:** 3
**Rating:** 6
**Confidence:** 3

**Summary:**

This paper proposed GMM-GDA, a graph data augmentation algorithm with better generalization abilities and faster training speed. GMM-GDA is presented based on a theoretical analysis relying a Rademacher complexity, which bounded the generalization error by the difference between the augmented data and original data. Furthermore, this paper verified the effectiveness of GMM-GDA from the perspective of influence functions and detailed experiments show the priority of the proposed algorithm.

**Strengths:**

1.	This paper analyzes the problem of GNN generalization capability, which is well-written and clearly clarified. The whole paper is easy to understand.

2.	This paper provides theoretical insights before presenting the algorithm.

3.	The experiments of this paper is closely related to the research goal proposed.

**Weaknesses:**

1.	This paper did not explain the necessity of performing data augmentation by GMM. The authors should strengthen the explanation of the relationship between theory and the algorithm.

2.	Some tables and figures do not seem to support the conclusions in the paper, which needs more explanation.

**Questions:**

1.	Based on theorem 3.1, the motivation of the proposed method is to guarantee the alignment of the augmented data and original data, so the authors apply GMM to fit the embeddings of the training data. But there lacks an explanation of why GMM is applied, it seems that a simple DNN (or other more complex generative data augmentation techniques) can also fit the embeddings. Please explain the advantages of GMM.

2.	It seems that only the post-readout function is trained by the combination of augmented data and original data (line 264). Why do not take several more iterations, and update the parameters of message passing layers? Please explain why only generate the embeddings of training data once but not re-generate them after updating the network.

3.	Figure 2 shows the influence scores of the augmented embeddings on different datasets. But the authors did not analyze why in dataset DD their algorithm perform worse. This is an interesting phenomenon and worth a deeper analysis.

4.	The authors claim that GMM-GDA is efficient in the augmentation steps and training steps and provide results in table 6 (line 315). But table 6 did not show the efficiency of GMM-GDA since it still cost much augmentation time or training time. Please explain why such a conclusion can be drawn from table 6.

5.	In the result of table 1&2, it seems that GMM-GDA has a better performance in the setting of GIN compared with GCN. This phenomenon worth a deeper analysis.

6.	The authors claim that the configuration models (line 470) is part of an ablation study, but it is hard to understand. Please explain more clearly about the conclusion of this experiment.

---

> ### Author Response · Authors · 2024-11-23
> **Response to Reviewer YuEk (1/2)**
>
> *We thank Reviewer YuEk for the feedback. In what follows, we address the raised questions and weaknesses point-by-point.*
>
> ***[Question 1 + Weaknesses 1] the Use of GMM Over Alternative Methods***
>
> Indeed, other generative techniques, such as generative models, could also be used to fit the embeddings of the training data. However, we chose to use GMMs for several important reasons, particularly the efficiency and effectiveness of GMMs in this context.
>
> GMMs are universal approximators, meaning they can effectively approximate any distribution, including the distribution of the graph embeddings. This property ensures that the augmented data is drawn from a distribution that closely aligns with the original data distribution, as shown in Theorem 3.1. While other generative methods could be used to fit the embeddings, GMMs have the advantage of being relatively simple and efficient in terms of computation (we can fit a GMM and generate new samples in very few secons). Unlike more complex methods such as generative models, GMMs can achieve high-quality approximations with minimal computational overhead.
>
> ***[Question 2] Distribution invariance in GMM-GDA***
>
> The decision to limit retraining to the post-readout function after augmenting the dataset is motivated by several considerations: 1/  Before the readout function, the GNN generates embeddings at the node level. Directly learning a distribution at the node level is more complex as the number of nodes varies across graphs. In contrast, the graph-level representations output are fixed-dimensional embeddings which make it easy to learn their distribution with a GMM and sample new graph representations from it. 2/ Augmenting the data increases the size of the training set by adding new graph representations, i.e. augmented graphs. However, the post-readout function is a relatively small component of the GNN, typically consisting of a linear layer followed by a softmax. As a result, training this component on the augmented dataset is computationally efficient. In contrast, retraining the message-passing layers, which involve multiple neighborhood aggregation steps and operate at the node level, would significantly increase computational training time due to the larger dataset size.
>
> ***[Question 3] Analyzing the influence scores on DD Dataset***
>
> Our approach GMM-GDA, generally performs better than the baselines for the DD dataset across both GIN and GCN backbones, as demonstrated in Tables 1 and 2. Figure 2 highlights that GMM-GDA exhibits greater effectiveness, as reflected by positive influence scores, on GIN compared to GCN. This behavior is consistent not only with our method but also with the baselines, as most graph data augmentation strategies tend to enhance test accuracy more significantly for GIN than for GCN when applied on DD.
>
> ***[Weaknesse 2 + Question 4] The augmentation time and training time***
>
> In addition to outperforming the baselines on most datasets, our approach offers an advantage in terms of time complexity (cf. Table 6 in Appendix E). The training time of baseline models varies depending on the augmentation strategy used, specifically, whether it involves pairs of graphs or individual graphs. Even in cases where a graph augmentation has a low computational cost for some baselines, training can still be time-consuming as multiple augmented graphs are required to achieve satisfactory test accuracy. For instance, methods like DropEdge, DropNode, and SubMix, while computationally simple, require generating multiple augmented samples at each epoch, thereby increasing the overall training time. In contrast, GMM-GDA introduces a more efficient approach by generating only one augmented graph per training instance, which is reused across all epochs. This design ensures a balance between computational efficiency and augmentation effectiveness, reducing the overall training burden while maintaining strong performance.
>
> The only baseline that is more time-efficient than our approach is GeoMix; however, our method consistently outperforms GeoMix across all settings, as shown in Tables 1 and 2.

---

> > ### Author Response · Authors · 2024-11-23
> > **Response to Reviewer YuEk (2/2)**
> >
> > ***[Question 5] Insights into GMM-GDA's Results with GIN and GCN***
> >
> > As mentioned in the paper, the comparison between different data augmentation approaches varies when using GCN and GIN. This is further explained by Theorem 3.3, which leverages influence functions to highlight how augmentation strategies may behave differently with model architectures. .
> >
> > Therefore, the observation that GMM-GDA performs better with GIN than GCN in few cases, can be explained by the  architectural differences between these models. GIN, with its stronger expressive power, closely approximates the Weisfeiler-Lehman graph isomorphism test, enabling it to better use the diversity of augmented graph representations generated by GMM-GDA. Despite these variations, our approach consistently outperforms the baselines in most cases
> >
> > Additionally, our method is not only more effective in improving performance but also demonstrates superior time efficiency. Unlike many baselines, GMM-GDA generates augmented data with minimal computational cost.
> >
> >
> > ***[Question 6] Configuration models based data augmentation***
> >
> > To address this, we have added the detailed steps of the configuration models-based data augmentation in Appendix C. This should provide a clearer understanding of the methodology of this experiment.

---

### Official Review · Reviewer_25Li · 2024-11-05

**Soundness:** 2
**Presentation:** 2
**Contribution:** 3
**Rating:** 5
**Confidence:** 3

**Summary:**

In this paper the authors introduce a method to perform data augmentation algorithm for graph datasets.
The algorithm leverages Gaussian Mixture Models (GMM) to find the maximum likelihood estimates for each cluster given by the embedding ofdifferent classes. Finlly they use the GMM to generate augmented data. Notice that augmented samples are generated directly in. the embedding space. The authors provide a bound for the Rademacher complexity fro the loss function  modified to account for augmented data.

**Strengths:**

- The proposed approach introduces a novel method for graph data augmentation.
- The problem studied is relevant and interesting
- The algorithm's time complexity is  analyzed.

**Weaknesses:**

- The clarity of the paper, particularly in Section 3, could be enhanced to guide the reader more effectively through the discussion.
- The method requires pre-training of the model  and moreover is dependent on the specific model architecture, meaning that the augmented dataset cannot be used by other GNN models.
- Baselines in Table 1 could be expanded to include additional augmentation techniques, such as edge insertion and feature drop.
- The metric used to measure the distance between $\mathcal{G}^{\lambda}$ and $\mathcal{G}$ in Theorem 3.1 and subsequent sections is not clearly defined.

**Questions:**

- In the case of GMM what does the parameter $\lambda$ represent ?
- Does the GMM-based sample generation method ensure that the augmented samples remain within the distribution of the original data?
 - Can the method be extended to different tasks on graphs? such as node classification and link prediction?
Comment:
If the maximum of the expectations $\mathbb{E}_{\lambda}[\mathcal{G}_n^{\lambda} - \mathcal{G}n]$ is non-zero, the Rademacher complexity of $\ell{aug}$ may not necessarily be smaller than the Rademacher complexity of $\ell$.

---

> ### Author Response · Authors · 2024-11-23
> **Response to Reviewer 25Li (1/2)**
>
> We thank Reviewer 25Li for the feedback. In what follows, we address the raised questions and weaknesses
> point-by-point.
>
> ***[Weaknesses 1] On the clarity of the paper***
>
> Thank you for your feedback. We acknowledge that the clarity of Section 3 could be improved to help guide the reader more effectively through the discussion. We have added more details in the revised manuscript. We will further enhance the clarity and organization of this section in the camera-ready version of the manuscript.
>
>  ***[Weaknesses 2] Pre-training of the model***
>
> To clarify, our approach does not require pre-training of the model in the traditional sense. Instead, we split the training process of the GNN into two distinct parts. The GNN consists of a sequence of message-passing layers followed by a shallow neural network, referred to as the post-readout function. We have provided a detailed summary of our approach in Algorithm 1.
>
> In our method, we first train the message-passing layers on the graph classification task. Then, we train the shallow post-readout network using the representations from both the original and augmented graphs. The second training phase, which involves the post-readout function, is very quick as it consists of a simple shallow MLP that can be trained in just a few seconds.
>
> As a result, while the total training time is increased due to this two-step process, the additional time required for training the second part is minimal. Therefore, the overall training time is not significantly impacted by our approach.
>
> ***[Weaknesses 2] Model architecture specificityl***
>
> We agree that the graph data augmentation method is dependent on the specific model architecture. The decision to perform augmentation at the level of graph representations, rather than directly on the graph inputs, is driven by Theorem 3.3. This theorem demonstrates that an effective data augmentation strategy should take into account the particular model architecture and its learned weights. Additionally, to evaluate a data augmentation strategy, it is necessary to train different GNN backbones separately on the augmented train set, e.g. evaluating a data augmentation approach on GCN and GIN requires training both GCN and GIN on the training set independently.
>
> ***[Weaknesses 3] Edge insertion and Deature drop Baselines***
>
> Thank you for suggesting additional baselines. We have already included baselines [1], which outperform these methods. Nevertheless, we will add the suggested baselines in the camera-ready version of the manuscript. While edge insertion is similar to the already-used DropEdge baseline, we will add it for more complete comparaison. However, feature drop cannot be applied across all datasets, as some lack node features, making this augmentation impractical in those cases.
>
> [1] Yoo, J., Shim, S., & Kang, U. (2022, April). Model-agnostic augmentation for accurate graph classification. In Proceedings of the ACM Web Conference 2022 (pp. 1281-1291).
>
> ***[Weaknesses 4] Metric in Theorem 3.1***
>
> The norm used to measure the distance between $\mathcal{G}^\lambda$ (the augmented graph) and $\mathcal{G}$ (the original graph) can correspond to any norm defined in the space of graphs, e.g. the spectral and Frobenius norms. The specific choice of the norm should align with the Lipschitz constant $L_{\text{Lip}}$. Importantly, the result of Theorem 3.1 remains unchanged regardless of the chosen norm because all norms are equivalent in the space of graphs.  If $L_{\text{Lip}}$ is taken to be the Lipschitz constant of the post-readout function only, then the norm corresponds to the difference between the graph-level embeddings produced by the readout function, i.e., $\|h_{\mathcal{G}^\lambda} - h_{\mathcal{G}}\|$.
>
> To address this, we explicitly clarify in the revised manuscript that the norm can be general norm that match the constant of $L_{\text{Lip}}$.
>
>
> ***[Question 1] Parameter $\lambda$ in the case of GMM***
>
> In our approach, the augmented hidden representations $h_{\mathcal{G}_{n}^{\lambda}}$  corresponds to a sampled vector from the GMM  distribution $\mathcal{P}_c$ that was previously fit on the hidden representations  $\mathcal{H}_c$ of the graphs in the training set with the same class  $c$.
>
> Formally, $h_{\mathcal{G}_{n}^{\lambda}}= A( \{\mathcal{H}_c\}_n, \lambda_c) = \lambda_c$, where $\lambda_c$ is sampled from the GMM distribution $\mathcal{P}_c$.

---

> > ### Author Response · Authors · 2024-11-23
> > **Response to Reviewer 25Li (2/2)**
> >
> > ***[Question 2] Distribution invariance in GMM-GDA***
> >
> > The GMM-based sample generation method ensures that the augmented samples remain within the distribution of the original data. This is because Gaussian Mixture Models (GMMs) are universal approximators, meaning they can approximate any probability distribution given a sufficient number of components. By learning the original data distribution with a GMM, the augmented samples generated from this model are drawn from a distribution that is very close to the original data distribution, preserving its key characteristics.
> >
> > Furthermore, the exponential decay in the Gaussian components of the GMM reduces the likelihood of sampling graph representations far from the learned distribution. This ensures that augmented samples stay consistent with the original data distribution.
> >
> >
> > ***[Question 3] Extension of the approach to other graph learning tasks***
> >
> > Unlike other baseline approaches, GMM-GDA is adaptable to data augmentation for tasks like node classification. The extension follows a similar framework, but here the focus shifts to learning the distribution at the node level rather than the graph level: 1/ We train the GNN on the node classification task. 2/ We use a Gaussian Mixture Model (GMM) to learn the distribution of node representations within the same class. 3/ We sample new node representations from the learned GMMs to augment the data.
> >
> > However, the method cannot be directly extended to the link prediction task. GMM-GDA is designed to learn and sample representations based on a specific feature space, either at the node or graph level, while for link prediction, the task depends on the pairwise relationships between node representations.
> >
> > ***[Comment] The Upper Bound in Theorem 3.1***
> >
> > It is true that if the maximum of the expectations $ \max_{n \in \{1, \ldots,N\}} \mathbb{E}_{\lambda \sim \mathcal{P}} \left [ \left \| \mathcal{G}_n^\lambda -\mathcal{G}_n \right \| \right ]$, is non-zero, the Rademacher complexity
> >
> > of $\ell_{aug}$   may not necessarily be smaller than the Rademacher complexity of $\ell$. However, by minimizing the term $ \mathbb{E}_{\lambda \sim \mathcal{P}} \left [ \left \| \mathcal{G}_n^\lambda -\mathcal{G}_n \right \| \right ]$],
> >
> > we are more likely to reduce the additional Rademacher complexity, thus increasing the chances of achieving a lower overall Rademacher complexity. Since we use Gaussian Mixture Models (GMMs) for augmenting the graph data, which are universal approximators, we ensure that the expectations $ \max_{n \in \{1, \ldots,N\}} \mathbb{E}_{\lambda \sim \mathcal{P}} \left [ \left \| \mathcal{G}_n^\lambda -\mathcal{G}_n \right \| \right ]$ approach zero.

---

> ### Comment · Reviewer_25Li · 2024-11-26
> **Response to authors' feedback**
>
> Regarding W4:
> The setting is clear when the distance between two graphs is defined in the embedding space, when not how it is defined? is it based on the Frobenius or spectral norm of the difference of their adjacency matrices?
>
> Regarding Q2:
> Even if GMMs are universal approximators, it is not guaranteed that the samples will remain within the distribution. Firstly, the approximation capabilities depend on the number of Gaussian components used. Secondly, the EM algorithm can converge to incorrect solutions, there are even simple cases in which it gets stuck in a local minimum.

---

> ### Author Response · Authors · 2024-11-28
> **Response to Reviewer 25Li (1/3)**
>
> We thank Reviewer 25Li very much for their follow-up questions. In what follows, we try to further clarify the remaining questions.
>
> ***[Weakness 4] Definition of Graph Distance in Non-Euclidean Spaces***
>
> Yes, the distance between two graphs can be based on the norm of the difference of their adjacency matrices, and the norm could be for example the Frobenius or spectral norm. As mentioned in our response, the inequality holds for both norms since all norms are equivalent in finite-dimensional spaces. Specifically, let us consider the graph space $(\mathbb{G}, \lVert \cdot \rVert_{\mathbb{G}})$ and the feature space $(\mathbb{X}, \lVert \cdot \rVert_{\mathbb{X}})$, where $\lVert \cdot \rVert_{\mathbb{G}}$ and $\lVert \cdot \rVert_{\mathbb{X}}$ denote the norms applied to the graph structure and features, respectively. Assuming a maximum number of nodes per graph, which is a realistic assumption for real-world data, the product space $\mathbb{G} \times \mathbb{X}$ is a finite-dimensional real vector space, and all the norms are equivalent. Thus, the choice of norm does not affect the theorem, as long as the Lipschitz constant is adjusted accordingly.
>
>
>  When considering only structural changes, with fixed node features, the distance between two graphs $\mathcal{G}^\lambda,\mathcal{G}$ is defined as:
>   $$\lVert\mathcal{G}^\lambda - \mathcal{G} \rVert= \lVert A - A^\lambda \rVert_{\mathbb{G}}, \qquad \qquad (1)$$ where $A,A^\lambda$ are respectively the adjacency matrix of $\mathcal{G}^\lambda,\mathcal{G},$ and the norm  $\lVert\cdot \rVert_{\mathbb{G}}$ can be the Frobenius or spectral norm. If both structural and feature changes are considered, the distance extends to:
>  $$\lVert\mathcal{G}^\lambda - \mathcal{G} \rVert= \alpha \lVert A - A^\lambda \rVert_{\mathbb{G}} +\beta \lVert X - X^\lambda \rVert_{\mathbb{X}}, \qquad \qquad (2) $$ where $X^\lambda, X$ are the node feature matrices of $\mathcal{G}^\lambda,\mathcal{G}$ respectively, and $\alpha, \beta$ are hyperparameters controlling the contribution of structural and feature differences, respectively.
>
> In most baselines graph augmentation techniques, such as for instance $\mathcal{G}$-Mixup, SubMix, and DropNode, the alignment between nodes in the original graph $\mathcal{G}$ and the augmented graph $\mathcal{G}^\lambda$ is known.  However, in cases where the node alignment is unknown, we must take into account the node permutations. The distance between the two graphs is then defined as:
> \begin{equation*}
>   \lVert\mathcal{G}^\lambda - \mathcal{G} \rVert= \min_{P \in \Pi} \left( \alpha \lVert A - P A^\lambda P^T \rVert_{\mathbb{G}}  + \beta \lVert X - P X^\lambda \rVert_{\mathbb{X}} \right), \qquad \qquad (3)
> \end{equation*}
> where $\Pi$ is the set of permutation matrices. The matrix $P$ corresponds to a permutation matrix used to order nodes from different graphs. By using Optimal Transport, we find the minimum distance over the set of permutation matrices, which corresponds to the optimal matching between nodes in the two graphs. This formulation represents the general case of graph distance, which has been used in the literature [1].
>
> Importantly, Theorem 3.1 applies to both Frobenius and spectral norms in the three scenarios (1), (2) and (3) that we lay out in our answer. Hence, we tried to state the theorem in full generality in our manuscript. But we appreciate that the current statement may be perceived to lack specificity. We have provided more details and insights on the norms for which Theorem 3.1 applies to in the revised version of the manuscript (in Section 3.1 Lines 229-35 and in Appendix G).
>
> [1] Abbahaddou, Y., Ennadir, S., Lutzeyer, J. F., Vazirgiannis, M., \& Boström, H. (2024)."Bounding the Expected Robustness of Graph Neural Networks Subject to Node Feature Attacks." In The Twelfth International Conference on Learning Representations.

---

> > ### Author Response · Authors · 2024-11-28
> > **Response to Reviewer 25Li (2/3)**
> >
> > ***[Question 2] GMM: Approximation and Convergence Challenges***
> >
> > It is true that we could have used other methods to model the distribution of the training graph data, such as generative models or alternative techniques to fit the GMM. However, the reasons for choosing GMM with the EM algorithm are three-fold:
> >
> >   1.***Superior Performance:***  Despite its simplicity, GMM with the EM algorithm consistently outperforms baseline methods across most cases in our experiments.
> >
> >   2.***Approximation Capability:*** By increasing $K$, a GMM can approximate any smooth target density. This property is a well-established result in statistical learning theory, as discussed in works like Deep Learning by [2]. In our approach, we ensure that a sufficiently large number of Gaussian components $K$ is used to accurately model the distribution of graph embeddings (by grid searching over the number of components from the values $\{2, 5, 10, 20, 30, 40, 50\}$), allowing us to approximate the underlying data distribution effectively.
> >
> >   3.***Computational Efficiency:*** The EM algorithm is computationally efficient, and allows us to generate augmented data at a relatively small computational cost, as highlighted in Section 3.3 of the paper.  The complexity to fit a GMM in $T$ iterations is $\mathcal{O}(N \cdot K\cdot T\cdot d^2)$ ($N$ is the number of points, $K$ is the number of Gaussians, and $d$ is the dimension of the features) [8]. Moreover, GMM is relatively easy to sample from, as highlighted in [3].
> >
> > To further validate empirically the choice of using GMMs fitted with the EM algorithm, we have explored an alternative approach to fitting the GMM using Variational Bayesian Inference (VBI) when using GCNs, presented as an ablation study in Appendix D of the originally submitted manuscript. The results demonstrate comparable performance to the EM algorithm, further validating the robustness of our approach.
> >
> > In reaction to your follow-up question, we have expanded our evaluation to include additional methods for modeling the distribution of the graph representations to provide additional comparison and motivate the use of GMMs with the EM algorithm. Specifically, our comparison now includes:
> >
> >   -***GMM w/ Variational Bayesian Inference (VBI):*** We specifically compared the Expectation-Maximization (EM) algorithm, discussed in the main paper, with the Variational Bayesian (VB) estimation technique [6] for parameter estimation of each Gaussian Mixture Model (GMM). The objective of including this baseline is to explore alternative approaches for fitting GMMs to the graph representations. In addition to the experimental results of VBI originally presented in Appendix D for the GCN backbone, we have extended the comparison between VBI and the EM algorithm to the GIN architecture as well.
> >
> >   -***Kernel Density Estimation (KDE) [4]:*** KDE is a Neighbor-Based Method and a non-parametric approach to estimating the probability density. KDE estimates the probability density function by placing a kernel function (e.g., Gaussian) at each data point. The sum of these kernels approximates the underlying distribution. Sampling can be done using techniques like Metropolis-Hastings. The purpose of using KDE as a baseline is to evaluate alternative distribution different from the Gaussian Mixture Model (GMM).
> >
> >   -***Copula-Based Methods [5]:***  We model the dependence structure between variables using copulas, while marginal distributions are modeled separately. We sample from marginal distributions and then transform them using the copula.
> >
> >   -***Generative Adversarial Network (GAN):*** GANs are powerful generative models that learn to approximate the data distribution through an adversarial process between two neural networks. To evaluate the performance of deep learning-based generative approaches for modeling graph representations, we included tGAN, a GAN architecture specifically designed for tabular data [7]. We particularly train tGAN on the graph representations and then sample new graph representation from the generator.

---

> ### Author Response · Authors · 2024-11-28
> **Response to Reviewer 25Li (3/3)**
>
> ***Table 1: Ablation Study GIN***
>
>
> | Model      | IMDB-BINARY           | IMDB-MULTI            | MUTAG                 | PROTEINS              | DD                    |
> |------------|-----------------------|-----------------------|-----------------------|-----------------------|-----------------------|
> | GMM w/ EM  | ***71.70 (4.24)*** | ***49.20 (2.06)*** | ***88.83 (5.02)*** | ***71.33 (5.04)*** | ***68.61 (4.62)*** |
> | GMM w/ VBI | 71.40 (2.65)          | 47.80 (2.22)          | 88.30 (5.19)          | 70.25 (4.65)          | 67.82 (4.96)          |
> | KDE        | 69.10 (3.93)          | 41.46 (3.02)          | 77.60 (6.83)          | 60.37 (3.04)          | 67.48 (6.18)          |
> | Copula     | 70.60 (2.61)          | 47.60 (2.29)          | 88.30 (5.19)          | 70.16 (4.55)          | 67.91 (4.90)          |
> | GAN        | 70.50 (3.80)          | 48.40 (1.71)          | ***88.83 (5.02)*** | ***71.33 (5.55)*** | 67.74 (4.82)          |
>
>
>
> ***Table 2: Ablation Study GCN***
>
> | Model      | IMDB-BINARY            | IMDB-MULTI            | MUTAG                 | PROTEINS              | DD                    |
> |------------|------------------------|-----------------------|-----------------------|-----------------------|-----------------------|
> | GMM w/ EM  | ***71.00 (4.40)*** | ***49.82 (4.26)*** | ***76.05 (6.47)*** | ***70.97 (5.07)*** | ***71.90 (2.81)*** |
> | GMM w/ VBI | ***71.00 (4.21)*** | 49.53 (4.26)          | ***76.05 (6.47)*** | ***70.97 (4.52)*** | 71.64 (2.90)          |
> | KDE        | 55.90 (10.29)          | 39.53 (2.87)          | 66.64 (6.79)          | 59.56 (2.62)          | 58.66 (3.97)          |
> | Copula     | 69.80 (4.04)           | 47.13 (3.45)          | 74.44 (6.26)          | 65.04 (3.37)          | 65.70 (3.04)          |
> | GAN        | 70.60 (3.41)           | 48.80 (5.51)          | 75.52 (4.96)          | 69.98 (5.46)          | 66.26 (3.72)          |
>
>
> We compare these approaches for both GCN and GIN in the table above. As noticed, GMM with EM consistently outperforms the alternative methods across most datasets in terms of accuracy. The VBI method, an alternative approach for estimating GMM parameters, yields comparable
> performance to the EM algorithm. This consistency across datasets highlights the effectiveness and robustness of GMMs in capturing the underlying data distribution.
>
> In certain cases, particularly with the GIN model, we observed competitive performance from the GAN approach, which, unlike GMM, requires additional training.  Hence, GMMs provide a more straightforward and efficient solution.
>
> We have included this detailed ablation study in the new version of the manuscript, c.f. Appendix D.
>
> [2] Ian Goodfellow, Yoshua Bengio, and Aaron Courville. Deep learning. MIT press, 2016.
>
> [3] Guo, H., Lu, C., Bao, F., Pang, T., Yan, S., Du, C., |\& Li, C. (2024). Gaussian mixture solvers for diffusion models. Advances in Neural Information Processing Systems, 36.
>
> [4] Härdle, W., Werwatz, A., Müller, M., Sperlich, S., Härdle, W., Werwatz, A., et al. \& Sperlich, S. (2004). Nonparametric density estimation. Nonparametric and semiparametric models, 39-83.
>
> [5] Roger B. Nelsen (1999), "An Introduction to Copulas", Springer. ISBN 978-0-387-98623-4
>
>
> [6] Dimitris G Tzikas, Aristidis C Likas, and Nikolaos P Galatsanos. (2008).The variational approximation for bayesian inference. IEEE Signal Processing Magazine, 25(6):131–146,
>
> [7] Miin-Shen Yang, Chien-Yo Lai, and Chih-Ying Lin. A robust em clustering algorithm for gaussian
> mixture models. Pattern Recognition, 45(11):3950–3961, 2012.
>
> [8] Xu, L., \& Veeramachaneni, K. (2018). Synthesizing tabular data using generative adversarial networks. arXiv preprint arXiv:1811.11264.
>
> ***[Question 2] Consistency with the Number of Gaussian Components***
>
> We experimented with values of \(K\) in the range from 2 to 50, finding that \(K=50\) was sufficient to balance model complexity and computational feasibility. In many cases, even a small number of components (e.g., \(K=2\) or \(K=5\)) yielded competitive performance (see Table 9 in the submitted manuscript). Below, we include a hyperparameter sensitivity analysis conducted on the GIN backbone using the IMDB-BINARY dataset, where we observed consistent results.
>
>
>
> |    | K=10        | K=20        | K=30        | K=40        | K=50        |
> |-----|-------------|-------------|-------------|-------------|-------------|
> | GIN | 71.1 (2.70) | 71.3 (2.28) | 71.3 (2.57) | 71.6 (2.72) | 71.7 (4.24) |

---

> > ### Author Response · Authors · 2024-12-01
> >
> > Dear Reviewer 25Li,
> >
> > We want to thank you again for the follow-up questions that allowed us to extend our ablation studies and to clarify our paper. As the discussion period ends soon, we would greatly appreciate a response if we have satisfactorily addressed your questions and concerns. If you have any further questions/concerns, please let us know and we will be happy to provide further answers.
> >
> > Best regards,
> > The Authors

---

### Official Review · Reviewer_7B8K · 2024-11-05

**Soundness:** 3
**Presentation:** 3
**Contribution:** 1
**Rating:** 1
**Confidence:** 5

**Summary:**

The paper discusses data augmentation for graphs. The concrete proposal is to use a Gaussian mixture model. The justification for this proposed approach is that Gaussian mixture models are universal density estimators.

**Strengths:**

Data augmentation for graphs is a topic that warrants investigation. There is little work on the subject.
Numerical evaluations are comprehensive

**Weaknesses:**

The proposed data augmentation method is not specific to graphs. It could apply to any data type. No arguments are given as to whether this is suitable way of augmenting graph datasets.

Numerical evaluations are comprehensive but underwhelming. Improvements are marginal relative to training without data augmentation. All but 1 improvement in Tables 1 and 2 are well within one standard deviation and can be explained by random chance.

**Questions:**

I do not understand Theorem 1. Please expand explanation.

---

> ### Author Response · Authors · 2024-11-22
> **Response to Reviewer 7B8K**
>
> *We thank Reviewer  7B8Ku for the feedback. In what follows, we address the raised questions and weaknesses point-by-point.*
>
> ***[Weaknesses 1] Graph-Specificity of our Approach***
>
> We acknowledge that our proposed data augmentation method is not inherently specific to graphs and could indeed be applied to other data types. However, we view this as a significant strength. Working on the field of Graph Machine Learning, we chose to apply this method to Graph Neural Networks (GNNs) because of its potential to enhance the performance of graph-based models. Moreover, several established graph augmentation methods, such as SubMix, GeoMix, and GraphMix, are adaptations of general data augmentation strategies originally designed for other domains, such as Mixup. These approaches have proven to be effective in graph contexts despite their general origins.
>
> Notably, the choice of augmenting the train dataset at the level of graph representation, and not at the level of graph input, is motivated by Theorem 3.3, which shows that a data augmentation strategy should depend on the specific model architecture and its weights.
>
> Another key motivation is that, unlike many existing baselines which focus on augmenting either the graph structure (e.g., perturbing edges) or the node features separately, our method operates directly on graph representations. This enables us to simultaneously augment both structural and feature information, leading to improved generalization in GNNs.
>
> ***[Weaknesses 2] Improvements of our Approach***
>
> We trained all baseline models using the same train/validation/test splits, GNN architectures, and hyperparameters to ensure a fair comparison. It is worth noting that the baselines also exhibit high standard deviations, which is a common characteristic in graph classification tasks. Unlike node classification, graph classification is known to  have larger variance in performance metrics [1].
>
> In addition to outperforming the baselines on most datasets, our approach offers an advantage in terms of time complexity (cf. Table 6 in Appendix E). The training time of baseline models varies depending on the augmentation strategy used, specifically, whether it involves pairs of graphs or individual graphs. Even in cases where a graph augmentation has a low computational cost for some baselines, training can still be time-consuming as multiple augmented graphs are required to achieve satisfactory test accuracy. For instance, methods like DropEdge, DropNode, and SubMix, while computationally simple, require generating multiple augmented samples at each epoch, thereby increasing the overall training time. In contrast, GMM-GDA introduces a more efficient approach by generating only one augmented graph per training instance, which is reused across all epochs. This design ensures a balance between computational efficiency and augmentation effectiveness, reducing the overall training burden while maintaining strong performance.
>
> [1] Bianchi, F. M., Lachi, V. (2024). The expressive power of pooling in graph neural networks. Advances in neural information processing systems, 36.
>
> ***[Question 1] Explanation of Theorem 3.1.***
>
> In Theorem 3.1, we established a mathematical framework to connect graph data augmentation with its impact on the generalization of GNNs. To evaluate GNN generalization, we employed the concept of Rademacher Complexity to derive a regret bound on the generalization gap. Rademacher Complexity, a fundamental concept in statistical learning theory, quantifies a model's capacity to generalize to unseen data by assessing its ability to fit random labels or noise. Better generalization is associated with a lower Rademacher Complexity. Theorem 3.1 demonstrates that data augmentation can reduce Rademacher Complexity, i.e., $\mathcal{R}(\ell_{aug}) \leq \mathcal{R}(\ell)$.  Specifically, the augmented graph should achieve a lower upper bound value for the following expression: $\max_{n \in \{1, \ldots,N\}}    \mathbb{E}_{\lambda \sim \mathcal{P}} \left [ \left \| \mathcal{G}_n^\lambda -\mathcal{G}_n \right \| \right ]$ which implies that the augmented data must follow the same distribution as the original graph representation. This condition is ensured by Gaussian Mixture Models (GMMs), which are universal approximators.
>
>
> *Given the brevity of your review, we hope that our additional explanations have helped you understand our work more deeply and give you the opportunity to reevaluate your assessment of our work.*

---

### Meta-Review · Area_Chair_18dn · 2024-12-24

**Metareview:**

This paper proposes a data augmentation method to enhance GNN generalization. The proposed method applied Gaussian Mixture Models (GMMs) to the readout features, and augment data following the trained GMM. Then the classifier is trained on the augmented data.

Unfortunately, the proposed method does not give a significant improvement of the performance. The proposed method just applies a general data augmentation method to the GNN setting and does not make use of any specific property of GNNs, which limits the novelty of the proposal. The reason why the proposed method does not give significant performance improvement would be that the method requires to fit GMMs (generative model) that is harder than just solving classification problem. Thus, this data augmentation approach does not necessarily perform well in not only GNNs but also general classification problems.

For these reasons, I don't recommend acceptance of this paper.

**Additional Comments On Reviewer Discussion:**

The drawbacks of this paper was not completely resolved through discussions. Indeed, the performance improvement looks just a chance rate.

---

### Decision · Program_Chairs · 2025-01-22

Reject